# Causal Identification for Complex Functional Longitudinal Studies

**Andrew Ying**
Irvine, CA 92606, USA
aying9339@gmail.com

## Abstract

Real-time monitoring in modern medical research introduces functional longitudinal data, characterized by continuous-time measurements of outcomes, treatments, and confounders. This complexity leads to uncountably infinite treatment-confounder feedbacks and infinite-dimensional data, which traditional causal inference methodologies cannot handle. Inspired by the coarsened data framework, we adopt stochastic process theory, measure theory, and net convergence to propose a nonparametric causal identification framework. This framework generalizes classical g-computation, inverse probability weighting, and doubly robust formulas, accommodating time-varying outcomes subject to mortality and censoring for functional longitudinal data. We examine our framework through Monte Carlo simulations. Our approach addresses significant gaps in current methodologies, providing a solution for functional longitudinal data and paving the way for future estimation work in this domain.

## 1 Introduction

The advent of real-time monitoring technologies in healthcare has led to the continuous-time measurement of outcomes, treatments, and confounders, which we term "functional longitudinal data." Here by "functional" we mean the first-generation data described in Wang et al. (2016), or termed curve data (Gasser et al., 1984; Rice & Silverman, 1991; Gasser & Kneip, 1995), operating over time. For example, the Medical Information Mart for Intensive Care IV (MIMIC-IV) (Johnson et al., 2023) is a freely accessible electronic health record (EHR) database that records ICU care data, including physiological measurements, laboratory values, medication administration, and clinical events. Another example is Continuous Glucose Monitoring (CGM) (Rodbard, 2016; Klonoff et al., 2017), an increasingly adopted technology for insulin-requiring patients that provides insights into glycemic fluctuations. CGM offers a real-time, high-resolution stream of data, capturing the intricate fluctuations in interstitial fluid glucose levels every few minutes.

These examples illustrate the recent prevalence of functional longitudinal data, highlighting the necessity of a causal framework, as understanding treatment effects is of paramount interest in these settings. However, there is a great lack of investigation of causal inference at the intersection of longitudinal data and functional data. Even identifying causal parameters of interest through observed data becomes highly nontrivial in this setting, due to the issue of uncountably infinite treatment-confounder feedbacks (Hernán & Robins, 2020) within functional longitudinal data. Treatment-confounder feedbacks occur when treatments taken over time influence variables (confounders) that in turn affect future treatments. For example, in a medical study, a patient's current medication (treatment) could affect their future health (a confounder), and their health might determine which medications they receive later. This back-and-forth interaction over time creates a cycle that is difficult to disentangle when analyzing causal relationships. In functional longitudinal data, this feedback becomes even more complex because both treatments and confounders are recorded as continuous functions over time rather than at discrete time points.

Moreover, functional longitudinal data, modeled as infinite-dimensional, continuous-time stochastic processes, demand a measure-theoretic foundation to ensure mathematical rigor. This requirement introduces complexities far beyond the scope of classical causal inference, which typically assumes finite-dimensional data and more elementary statistical tools. Apart from the mathematical rigor,

from a statistical level, traditional approaches for handling functional data often rely on parametric or semi-parametric modeling assumptions, such as smoothness or sparsity, to facilitate analysis and reduce dimensionality. However, these assumptions are typically made for mathematical convenience rather than being grounded in prior knowledge. As a result, inferences drawn from such models may reflect the assumptions as much as, or more than, the data itself.

To bridge this gap, we aim to propose a novel identification framework for functional longitudinal data with time-varying outcomes subject to mortality and censoring, who enjoys the nonparametric property, making it more flexible and adaptable to various datasets.

We first define a causal quantity representing the mean of counterfactual outcomes under an idealized randomized world. To connect the observed data distribution to this idealized world, inspired by the coarsened data framework (Heitjan & Rubin, 1991) and through the application of continuous-time stochastic process theory and measure theory, we upgrade classical causal assumptions to accommodate functional longitudinal data nonparametrically. These together resolve the issue of uncountably infinite treatment-confounder feedbacks (Hernán & Robins, 2020) for functional longitudinal data. We generalize the well-known g-computation formula, inverse probability weighting formula, and double robust formula. We examine our identification framework through Monte Carlo simulations.

The paper is organized as follows. In Section 2 we present a literature review of related work. In Section 3, we define the notation and parameters of interest. Then we propose identification assumptions and generalize the well-known g-computation, inverse probability weighting, and double robust formulas (Hernán & Robins, 2020). Additionally, we prove that our identification is nonparametric. We conduct Monte Carlo simulations to examine our framework in Section 4. Section 5 discusses future directions. While this paper builds a population-level framework with numerical results, it does not explore estimation or associated inference, which is beyond the scope of this study and left for future research.

## 2 RELATED WORK

**Causal Inference for Non-Functional Longitudinal Studies.** Current causal frameworks for longitudinal studies fall into two main categories: "regular longitudinal studies," where time advances in fixed intervals (Greenland & Robins, 1986; Robins, 1986), and "irregular longitudinal studies," where events occur at random but discrete time points (Lok, 2008; Røysland, 2011; Rytgaard et al., 2022). "Regular longitudinal studies" are straightforward but limited to structured designs, while "irregular longitudinal studies," such as those by Rytgaard et al. (2022), accommodate random visit times by modeling treatment and confounder processes as counting processes. These approaches assume finite treatment-confounder feedbacks and finite-dimensional data, relying on stepwise paths and joint densities for causal identification.

However, modern medical studies often generate functional longitudinal data through continuous monitoring of treatments and confounders, as seen in intensive care settings (Johnson et al., 2016; 2018) and wearable devices for chronic disease management (Mastrototaro, 2000; Klonoff, 2005; Rodbard, 2016). Existing frameworks, designed for discrete-time or stepwise processes, are insufficient for such infinite-dimensional data, highlighting the need for new causal inference tools that accommodate the complexities of functional longitudinal data.

**Causal Inference for Functional Data.** Existing research on causal inference has examined functional data within observational studies, as highlighted in works by (Miao et al., 2020; Zhang et al., 2021; Tan et al., 2022). These studies share a similar data format with our analysis. However, our approach distinguishes itself by focusing on the time-dependent nature of longitudinal studies, where data evolve continuously over time. In contrast, the cited works primarily address "point exposure," which looks at the impact of a single treatment or covariates measured at beginning of a study, without accounting for how treatments or covariates may change and interact over a longer period.

**Existing Work for Functional Longitudinal Data.** The only exceptions that investigated causal inference for functional longitudinal data are Ying (2024a) and Sun & Crawford (2022). However, Ying (2024a) only investigated a single outcome, measured at the end of some medical studies, neither proving the nonparametric property nor conducting any numerical investigation. On the other hand, Sun & Crawford (2022) imposed stochastic differential equations with stringent parametric

assumptions. This situation highlights a significant gap in methodological advancements within the field. A related study (Ying, 2024b) explores a more general framework, building upon the methods and insights presented here. However, it does not include numerical examples or practical verifications, one of our contributions.

## 3 PROPOSED METHOD

### 3.1 PREPARATION

Consider a longitudinal study spanning from time 0 to $\infty$:

- $A(t)$ and $L(t)$ are two stochastic processes denoting the treatment administered and the measured confounders, respectively, at any given time $t$. At any time, $A(t)$ and $L(t)$ could be binary, categorical, continuous, or even functional. We denote $\bar{A}(t) = \{A(s) : 0 \leq s \leq t\}$ and $\bar{L}(t) = \{L(s) : 0 \leq s \leq t\}$, with $\bar{A}$ and $\bar{L}$ representing the collections of treatments and confounders over the entire study.

- We are interested in an outcome of interest $Y(t)$, as a subset of $L(t)$, that is, $Y(t) \subset L(t)$. This notation was chosen purely for simplicity. We are not assuming $Y(t)$ must affect treatment assignment but instead allow this dependency to exist or not. This flexibility is critical as in many cases (e.g., disease progression), outcomes can influence treatment adjustments, and therefore acting as a confounder as well.

- Let $T$ be a time-to-event endpoint, for instance, death, and $C$ be the right censoring time. Define $X = \min(T, C)$ as the censored event time and $\Delta = \mathbb{1}(T \leq C)$ the event indicator. Therefore when $\Delta = 1$, $X = T$ and when $\Delta = 0$, $X = C$. We also define $N(t) = \mathbb{1}(X \leq t)$ as the counting processes of $X$.

- Write the counterfactual time-to-event endpoint $T_{\bar{a}}$ and counterfactual covariates $L_{\bar{a}}(t)$, for any $\bar{a} \in \mathcal{A}$, where $\mathcal{A}$ encompasses all possible values of $\bar{a}$. Therefore we have $X_{\bar{a}} = \min(T_{\bar{a}}, C)$ and $\Delta_{\bar{a}} = \mathbb{1}(T_{\bar{a}} < C)$. We assume that the future cannot affect the past, that is, $\mathbb{1}(T_{\bar{a}} \geq t) = \mathbb{1}(T_{\bar{a}'} \geq t)$ and $L_{\bar{a}}(t) = L_{\bar{a}'}(t)$ whenever $\bar{a}(t) = \bar{a}'(t)$. We also write $T_{\mathcal{A}} = \{T_{\bar{a}} : \bar{a} \in \mathcal{A}\}$ and $\bar{L}_{\mathcal{A}} = \{\bar{L}_{\bar{a}} : \bar{a} \in \mathcal{A}\}$.

- The full data are $\{\bar{A}, C, T_{\mathcal{A}}, \bar{L}_{\mathcal{A}}\}$ and the observed data are $\{\bar{A}, X, \Delta, \bar{L}\}$. Note that on the observed data level, $A(t)$ and $L(t)$ are not observed for $t \leq X$ or defined for $t \leq T$. For easier notation in this paper, we offset $A(t) = A(X)$ and $\bar{L}(t) = L(X)$ for observed data whenever $t > X$. In this way, the stochastic processes $A(t)$ and $L(t)$ are well defined at any $t > 0$.

- Define $\mathscr{F}_t = \sigma(\{A(s), L(s), \mathbb{1}(X \leq s), \mathbb{1}(X \leq s)\Delta : \forall s \leq t\})$ as a filtration of information observed up to time $t$. Also we write $\mathscr{F}_{t-} = \sigma(\cup_{0 \leq s < t} \mathscr{F}_t)$ and $\mathscr{G}_t = \sigma(\{\mathscr{F}_{t-}, A(t)\})$. We define $\mathscr{G}_{\infty+} = \mathscr{F}_{\infty}$. We write $\mathscr{F}_{0-}$ and $\mathscr{G}_{0-}$ as the trivial sigma algebra for convenience. Note that $X$ is a stopping time with respect to $\mathscr{F}_t$, with $\mathscr{F}_{\infty} = \mathscr{F}_X = \sigma(\{\bar{A}, X, \Delta, \bar{L}\})$.

- We use $\mathbb{P}(\mathrm{d}x\mathrm{d}\delta\mathrm{d}\bar{a}\mathrm{d}\bar{l})$ (Bhattacharya & Waymire, 2007; Durrett, 2019; Gill & Robins, 2001) to represent the measure on the path space induced by the stochastic processes. Note that this is not a density function. [1]We use $\mathbb{E}$ as the corresponding expectation.

In the context of MIMIC-III, $A(t)$ could represent antibiotics usage at time $t$, and $L(t)$ may include a range of clinical measurements, such as severity of illness scores, vital signs, laboratory values, blood gas values, urine output, weight, height, age, gender, service type, total fluid intake, and total fluid output at time $t$. The outcome $Y(t)$ might measure illness progression influenced by antibiotics,

---

[1]Measure theory is essential in continuous-time stochastic processes because it addresses challenges that density-based approaches cannot handle. Many processes, such as those with jumps or irregular paths, lack well-defined densities, yet measure theory allows us to work directly with their distributions. Additionally, stochastic processes often evolve in infinite-dimensional spaces (e.g., path spaces), where defining densities is impractical, but measure-theoretic methods naturally extend. It also enables rigorous definitions of key concepts like conditional probabilities and expectations, which are foundational in this field. Beyond practicality, measure theory aligns with the tradition and standard methodology in stochastic process theory, making it both a necessary and convenient choice.

such as changes in severity scores over time. $T$ could represent the time to discharge or mortality, with $C$ as the time the patient is censored, such as at the end of data collection. Counterfactual outcomes like $T_{\bar{a}}$ might represent the time to recovery under a specific antibiotic regimen $\bar{a}$, and $L_{\bar{a}}(t)$ could represent the trajectory of severity scores under that treatment.

Similarly, in the context of CGM, $A(t)$ represents insulin dosage at time $t$, and $L(t)$ includes glucose levels and immediate behavioral changes such as diet, medications, and physical activity at time $t$. The outcome $Y(t)$ represents the glucose levels monitored in real time in response to insulin adjustments. $T$ could represent the time to a severe glucose event, with $C$ as the time the patient stops CGM usage. Counterfactual outcomes like $T_{\bar{a}}$ represent the time to stable glucose control under a specific insulin dosing regime $\bar{a}$, and $L_{\bar{a}}(t)$ captures the counterfactual glucose trajectory.

We are interested in learning a marginal mean of transformed potential outcomes including a time-to-event outcome and an outcome process under a user-specified treatment regime in the absence of censoring,

$$\int_{\mathcal{A}} \mathbb{E}(\nu(T_{\bar{a}}, \bar{Y}_{\bar{a}}))\mathbb{G}(\mathrm{d}\bar{a}), \tag{1}$$

where $\nu$ is some user-specified function and $\mathbb{G}$ is a priori defined (signed) measure on $\mathcal{A}$, representing a stochastic treatment regime. Here stochastic treatment regimes do not prescribe a specific treatment value but instead define the probability of receiving each possible treatment. In other words, it assigns treatments randomly according to a specified probability distribution. Stochastic treatment regimes offer a flexible approach for modeling treatments that are either continuous or challenging to precisely quantify. Unlike deterministic regimes, where treatment decisions are fixed, stochastic regimes introduce variability, enabling a broader range of real-world applications. This approach is particularly beneficial in scenarios where treatments are not strictly prescribed but instead follow probabilistic guidelines or are influenced by patient behavior or external factors. Examples of $\nu(\cdot)$:

- $\nu(T_{\bar{a}}, \bar{Y}_{\bar{a}}) = \mathbb{1}(T_{\bar{a}} > t)$, for some time $t > 0$, identifies the effect of $\bar{a}$ on the survival probability. For instance, in MIMIC-III, this could represent the probability of a patient surviving beyond time $t$ under a specific antibiotic regimen $\bar{a}$. Alternatively, $\nu(T_{\bar{a}}, \bar{Y}_{\bar{a}}) = \min(T_{\bar{a}}, \tau)$ represents the restricted mean survival time.

- $\nu(T_{\bar{a}}, \bar{Y}_{\bar{a}}) = Y_{\bar{a}}(\tau)$ is the outcome measured at time $\tau$, for some $\tau > 0$. In CGM, it could correspond to the glucose level at time $\tau$ under a specific insulin dosing strategy $\bar{a}$. Alternatively, $\nu(T_{\bar{a}}, \bar{Y}_{\bar{a}}) = Y_{\bar{a}}(T_{\bar{a}})$ represents the outcome measured at the time-to-event $T_{\bar{a}}$. In MIMIC-III, this might capture the severity of illness or lactate level at the time of recovery or death. Finally, $\nu(T_{\bar{a}}, \bar{Y}_{\bar{a}}) = \int_0^\tau w(t)Y_{\bar{a}}(t)dt/\tau$ represents the weighted averaged outcome over $[0, \tau]$, where $w(t)$ is a user-specified weight function.

We assume $\mathbb{E}(\nu(T_{\bar{a}}, \bar{Y}_{\bar{a}}))$ is integrable against $\mathbb{G}$. This exploration encompasses marginal means under static treatment regimes, as discussed in various literature (Rytgaard et al., 2022; Cain et al., 2010; Young et al., 2011; Hernán & Robins, 2020). This quantity can be seen as the mean of counterfactual outcomes under an idealized randomized world, where $\bar{a}$ is randomized to follow a stochastic treatment regime $\mathbb{G}$. Examples of $\mathbb{G}$:

- $\mathbb{G} = \mathbb{1}(\bar{A} = \bar{a})$ representing the averaged treatment outcome under a specific regime is of interest. $\mathbb{G} = \mathbb{1}(\bar{A} = \bar{a}) - \mathbb{1}(\bar{A} = \bar{a}')$ representing the averaged treatment effect of specific regime $\bar{a}$ versus another $\bar{a}'$.

- For treatments like physical activity, which is inherently variable and challenging to quantify precisely, can be modeled using stochastic regimes. For example, rather than prescribing a strict regimen of 30 minutes of exercise daily, a stochastic regime might increase the likelihood of patients engaging in activity based on encouragements or incentives. In both case, $\mathbb{G}$ can be specified as a distribution instead of delta masses.

## 3.2 IDENTIFICATION ASSUMPTIONS

We have defined the parameter of interest (1). Intuitively if treating treatment process $\bar{A}$ as a selection process (Heitjan & Rubin, 1991), (1) is the mean of $\nu(T_{\bar{a}}, \bar{Y}_{\bar{a}})$ when $\bar{A}$ were to follow $\mathbb{G}$ and there is no censoring. To create such a pseudo-population, note that for any sequences of partitions

$\{\Delta_K[0, \infty]\}_{K=1}^\infty$, where we have a partition $\Delta_K[0, \infty]$ over $[0, \infty]$ is a finite sequence of $K + 1$ numbers of the form $0 = t_0 < \cdots < t_K = \infty$, we loosely have the following decomposition

$$\mathbb{P}(\mathrm{d}x\mathrm{d}\delta\mathrm{d}\bar{a}\mathrm{d}\bar{l}) \tag{2}$$

$$= \prod_{j=0}^{K-1} F_T(t_{j+1}|\mathscr{F}_{t_j})^{\Delta(N(t_{j+1})-N(t_j))}(1 - F_T(t_{j+1}|\mathscr{F}_{t_j}))^{(1-\Delta)N(t_{j+1})} \tag{3}$$

$$F_C(t_{j+1}|\mathscr{F}_{t_j})^{(1-\Delta)(N(t_{j+1})-N(t_j))}(1 - F_C(t_{j+1}|\mathscr{F}_{t_j}))^{\Delta N(t_{j+1})} \tag{4}$$

$$\mathbb{P}(\mathrm{d}\bar{l}(t_{j+1)})|\mathscr{F}_{t_j})\,\mathbb{P}(\mathrm{d}\bar{a}(t_{j+1})|\mathscr{F}_{t_j})., \tag{5}$$

where we temporarily write $F_T$ and $F_C$ as the distribution functions of $T$ and $C$. We intervene treatment distribution at each time $t_j$ to approximate the pseudo-population where $\bar{A}$ were to follow $\mathbb{G}$ as:

$$\mathbb{P}_{\Delta_K[0,\infty],\mathbb{G}}(\mathrm{d}x\mathrm{d}\delta\mathrm{d}\bar{a}\mathrm{d}\bar{l}) \tag{6}$$

$$= \prod_{j=0}^{K-1} F_T(t_{j+1}|\mathscr{F}_{t_j})^{\Delta(N(t_{j+1})-N(t_j))}[1 - (1 - \Delta)N(t_{j+1})] \tag{7}$$

$$\mathbb{P}(\mathrm{d}\bar{l}(t_{j+1}))|\mathscr{F}_{t_j})\mathbb{G}(\mathrm{d}\bar{a}(t_{j+1})|\bar{a}(t_j)). \tag{8}$$

Here informally, one might understand this intervention as we replace the censoring distribution (4) and treatment distribution (5) between $(t_j, t_{j+1}]$ by no censoring as in (7) and targeted treatment distribution $\mathbb{G}$ as in (8). For readers unfamiliar with intervention-based causal inference language, we refer to Rytgaard et al. (2022, Definitions 1 & 2). A more formal and mathematically rigorous decompositions are given in Section A the appendix.

To eliminate confounder bias, we need to make sure there is no unmeasured confounders. We adapt the commonly known "coarsening at random" (Heitjan & Rubin, 1991) assumption into:

**Assumption 1** (Full conditional randomization). *The treatment assignment is independent of the all potential outcomes and covariates given history, in the sense that there exists a bounded function $\varepsilon(t, \eta) > 0$ with $\int_0^\infty \varepsilon(t, \eta)dt \to 0$ as $\eta \to 0$, such that for any $t \in [0, \infty]$, $\eta > 0$,*

$$\sup_{\bar{a} \in \mathcal{A}} \mathbb{E}(\| \mathbb{P}(\mathrm{d}t_{\bar{a}}\mathrm{d}\bar{l}_{\bar{a}}|\bar{A}(t + \eta), \mathscr{F}_t) - \mathbb{P}(\mathrm{d}t_{\bar{a}}\mathrm{d}\bar{l}_{\bar{a}}|\mathscr{F}_t)\|_{\mathrm{TV}}) < \varepsilon(t, \eta), \tag{9}$$

*where $\| \cdot \|_{\mathrm{TV}}$ is the total variation norm over the path space's signed measure space.*

This assumption claims that, the treatment distribution, or equally, the probability of coarsening, in a small period of time around $t$, only depends on the observed data up to time $t$ and independent of further part of counterfactuals. This assumption says in a approximating sense that there is no common cause between treatment decision between time $[t, t + \eta]$ and all future counterfactual confounders. Intuitively and unofficially, one might see this as saying $\mathbb{P}(\mathrm{d}t_{\bar{a}}\mathrm{d}\bar{l}_{\bar{a}}|\bar{A}(t + \eta), \mathscr{F}_t) \approx \mathbb{P}(\mathrm{d}t_{\bar{a}}\mathrm{d}\bar{l}_{\bar{a}}|\mathscr{F}_t)$, or approximately, $(T_{\bar{a}}, \bar{L}_{\bar{a}}) \perp \bar{A}(t + \eta)|\mathscr{F}_t$.

We also need an assumption over the censoring mechanism to eliminate the censoring bias. We consider the well-known conditionally independent censoring assumption (Tsiatis, 2006; Andersen et al., 2012). Define the full data censoring time hazard function as

$$\lambda_C(t|T, \bar{A}, \bar{L}) = \lim_{dt \to 0} \mathbb{P}(C \leq t + \mathrm{d}t|C > t, T, \bar{A}, \bar{L})/\mathrm{d}t. \tag{10}$$

The following assumption requires that the full data censoring time hazard at time $t$ only depends on the observed data up to time $t$.

**Assumption 2** (Conditional independent censoring). *The censoring mechanism is said to be conditionally independent if*

$$\lambda_C(t|T, \bar{A}, \bar{L}) = \lim_{dt \to 0} \mathbb{P}(C \leq t + \mathrm{d}t|C > t, T > t, \bar{A}(t), \bar{L}(t))\mathbb{1}(T > t)/\mathrm{d}t. \tag{11}$$

Note that in order to overcome the continuous-time issue, here we impose Assumption 1 over an infinitesimal period of time. This type of idea is also adopted in Assumption 2. Note that how Assumption 2 is given on the intensity process whereas Assumption 1 is on the conditioning event. This is because one does not have intensity process for a general stochastic process.

With Assumptions 1 and 2, we are able to show that whenever $|\Delta_K[0,\infty]| \to 0$, $\mathbb{P}_{\Delta_K[0,\infty],\mathbb{G}}$ approximates a pseudo-measure where treatment distribution are intervened by uncountable times into following $\mathbb{G}$, where the mesh $|\Delta_K[0,\infty]|$ of a partition $\Delta_K[0,\infty]$ is defined as $\max[\max_{i=0,\cdots,K-1}(t_{j+1}-t_j), 1/t_{K-1}]$, representing the maximum gap length of the partition:

**Proposition 1** (Intervenable). *Under Assumptions 1 and 2, the measures $\mathbb{P}_{\Delta_K[0,\infty],\mathbb{G}}$ converges to the same (signed) measure $\mathbb{P}_{\mathbb{G}} := \mathbb{P}(\mathrm{d}x_{\bar{a}}\mathrm{d}l_{\bar{a}})\mathbb{G}(\mathrm{d}\bar{a})\delta_{\bar{a}}$ in the total variation norm on the path space as the meshes shrink to zero, regardless of the choices of partitions, that is,*

$$\| \mathbb{P}_{\Delta_K[0,\infty],\mathbb{G}}(\mathrm{d}x\mathrm{d}\delta\mathrm{d}\bar{a}\mathrm{d}\bar{l}) - \mathbb{P}(\mathrm{d}x_{\bar{a}}\mathrm{d}\bar{l}_{\bar{a}})\mathbb{G}(\mathrm{d}\bar{a})\delta_{\bar{a}}\|_{\mathrm{TV}} \to 0, \tag{12}$$

*whenever $|\Delta_K[0,\infty]| \to 0$.*

We refer $\mathbb{P}_{\mathbb{G}}$ as the *target distribution*. This proposition has helped us to use the intervened observed data distribution $\mathbb{P}_{\Delta_K[0,\infty],\mathbb{G}}(\mathrm{d}x\mathrm{d}\delta\mathrm{d}\bar{a}\mathrm{d}\bar{l})$ to identify the target counterfactuals distribution $\mathbb{P}(\mathrm{d}x_{\bar{a}}\mathrm{d}\bar{l}_{\bar{a}})\mathbb{G}(\mathrm{d}\bar{a})\delta_{\bar{a}}$ in an asymptotic sense. The following assumption links the observed variable with the counterfactuals.

**Assumption 3** (Full consistency). *For any t,*

$$T = T_{\bar{A}}, L(t) = L_{\bar{A}}(t). \tag{13}$$

The full consistency assumption links the observed outcome and the potential outcome via the treatment actually received. It says that if an individual receives the treatment $\bar{A} = \bar{a}$, then his/her observed outcome $Y$ matches $Y_{\bar{a}}$.

The following assumption ensures that the observed data can identify the target distribution.

**Assumption 4** (Positivity).

$$\mathbb{P}_{\mathbb{G}} \ll \mathbb{P}. \tag{14}$$

With the above assumptions, we are able to generalize the well-known identification formulas: g-computation, inverse probability weighting, and double robust formulas, into functional longitudinal data. Note that our assumptions can be weaker but chosen for ease to interpret.

## 3.3 IDENTIFICATION FORMULAS

Below we show how we generalize the well-known g-computation, inverse probability weighting, and double robust formulas for "functional longitudinal data." For readers unfamiliar with the concepts, we refer to Hernán & Robins (2020). We also prepare a review for these formulas in discrete-time longitudinal data in Section B in the appendix.

**Definition 1** (G-computation process). *Under Assumptions 1 and 2, define*

$$H_{\mathbb{G}}(t) = \mathbb{E}_{\mathbb{G}}[\nu(X,\bar{Y})|\mathscr{G}_t], \tag{15}$$

*as a projection process, which is apparently a $\mathbb{P}_{\mathbb{G}}$-martingale. We call $H_{\mathbb{G}}(t)$ the g-computation process. Note that*

$$H_{\mathbb{G}}(\infty) = \nu(X,\bar{Y}), \quad H_{\mathbb{G}}(0-) = \mathbb{E}_{\mathbb{G}}[\nu(X,\bar{Y})]. \tag{16}$$

The g-computation process intuitively serves as a consecutive adjustment of the target $\nu(X,\bar{Y})$ from $\infty$ to 0. It represents a mix of original conditional distributions of covariate process together with the intervened treatment process $\mathbb{G}$, from end of study to the beginning. Following this adjustment to the beginning of study, we have:

**Theorem 1** (G-computation formula). *Under Assumptions 1, 2, 3, and 4, (1) is identified via a g-computation formula as*

$$\int_{\mathcal{A}} \mathbb{E}(\nu(T_{\bar{a}},\bar{Y}_{\bar{a}}))\mathbb{G}(\mathrm{d}\bar{a}) = H_{\mathbb{G}}(0-). \tag{17}$$

**Definition 2** (Inverse probability weighting process). *Under Assumptions 1 and 2, define*

$$Q_{\mathbb{G}}(t) = \mathbb{E}\left(\frac{\mathrm{d}\,\mathbb{P}_{\mathbb{G}}}{\mathrm{d}\,\mathbb{P}}\bigg|\mathscr{G}_t\right), \tag{18}$$

*as the Radon-Nikodym derivative at any time t, which is apparently a $\mathbb{P}$-martingale. We call $Q_{\mathbb{G}}(t)$ the inverse probability weighting process. Note that*

$$Q_{\mathbb{G}}(\infty) = \mathbb{E}\left(\frac{\mathrm{d}\,\mathbb{P}_{\mathbb{G}}}{\mathrm{d}\,\mathbb{P}}\bigg|\mathscr{G}_\infty\right) = \frac{\mathrm{d}\,\mathbb{P}_{\mathbb{G}}}{\mathrm{d}\,\mathbb{P}}, \quad Q_{\mathbb{G}}(0-) = 1. \tag{19}$$

The IPW process intuitively serves as a continuous adjustment of the treatment process $\bar{A}$ from 0 to $\infty$, using which as weights one may create a pseudo population as if the whole process were to follow $\mathbb{P}_{\mathbb{G}}$. It reweights the observed data distribution $\mathbb{P}$ into $\mathbb{P}_{\mathbb{G}}$ from the beginning of the study to the end. Following this reweighting throughout the longitudinal study, we have:

**Theorem 2** (Inverse probability weighting formula). *Under Assumptions 1, 2, 3, and 4, (1) is identified via an inverse probability weighting formula as*

$$\int_{\mathcal{A}} \mathbb{E}(\nu(T_{\bar{a}}, \bar{Y}_{\bar{a}}))\mathbb{G}(\mathrm{d}\bar{a}) = \mathbb{E}\left[Q_{\mathbb{G}}(\infty)\nu(X, \bar{Y})\right]. \tag{20}$$

For any two $\mathscr{G}_t$-adapted processes $H(t)$ and $Q(t)$, and a partition $\Delta_K[0, \infty]$, we write

$$\Xi_{\Delta_K[0,\infty]}(H, Q) = \sum_{j=0}^{K} Q(t_j)\left\{\int H(t_{j+1})\mathbb{G}(\mathrm{d}\bar{a}(t_{j+1})|\bar{A}(t_j)) - H(t_j)\right\} + \int H(0)\mathbb{G}(\mathrm{d}\bar{a}(0)). \tag{21}$$

We also define $\Xi(H, Q)$ as the limit of $\Xi_{\Delta_K[0,\infty]}(H, Q)$ in probability whenever it exists. We have:

**Theorem 3** (Doubly robust formula). *Under Assumptions 1, 2, 3, and 4, for any $\mathscr{G}_t$-adapted processes $H(t)$ and $Q(t)$ at the law where $\Xi(H, Q)$, as the limit of $\Xi_{\Delta_K[0,\infty]}(H, Q)$ in probability, exists and*

$$\lim_{|\Delta_K[0,\infty]|\to 0} \mathbb{E}(\Xi_{\Delta_K[0,\infty)}(H, Q)) = \mathbb{E}(\Xi(H, Q)), \tag{22}$$

*we have*

$$\int_{\mathcal{A}} \mathbb{E}(\nu(T_{\bar{a}}, \bar{Y}_{\bar{a}}))\mathbb{G}(\mathrm{d}\bar{a}) = \mathbb{E}(\Xi(H, Q)), \tag{23}$$

*provided that either $H = H_{\mathbb{G}}$ or $Q = Q_{\mathbb{G}}$.*

As one can see, the doubly robust formula provides extra protection against possible misspecification on either the g-computation process or the IPW process.

### 3.4 No restrictions on the observed data distribution: A Nonparametric Framework

For functional data, where the complexity of continuous, infinite-dimensional outcomes makes it even harder to justify any specific model, relying on parametric assumptions becomes especially unrealistic. In this subsection, we demonstrate that our identification framework imposes no restrictions on the observed data. Our framework deliberately separates modeling assumptions from identification, focusing purely on structural assumptions necessary for causal inference. This ensures that the framework extracts information only from the data, avoiding the risk of introducing unwarranted or misleading conclusions based on arbitrary assumptions. This property is advantageous for researchers and practitioners because nonparametric frameworks are flexible and require minimal assumptions, making them robust and adaptable to diverse datasets. This aligns with the recent assumption-lean efforts in the causal inference community (Vansteelandt & Dukes, 2022; Vansteelandt et al., 2024).

We demonstrate this by proving that, for any given observed data distribution, we can identify a sequence of full data distributions—each satisfying Assumptions 1, 2, 3, and 4—such that their corresponding distributions on the observed data closely approximate the initial observed data distribution. That is, we write the set of all observed data distribution as $\mathcal{P}$ and its subset satisfying Assumptions 1, 2, 3, and 4 as $\mathcal{M}$, then we show that $\mathcal{M}$ is a dense subset of $\mathcal{P}$ in the total variation norm.

Up to now, we have used $\mathbb{P}$ to represent both the distribution on the sample space and the path space. In this subsection, we use $\mathbb{P}$ to denote the distribution on the observed data $(\bar{A}, X, \Delta, \bar{L})$ and $\mathbb{P}^F$ to denote the distribution on the full data $(\bar{A}, C, T_{\mathcal{A}}, \bar{L}_{\mathcal{A}})$. We have

**Theorem 4.** *When the path space consists of all piece-wise continuous processes, for any measure $\mathbb{P}$ over the observed data $(\bar{A}, X, \Delta, \bar{L})$, there exists a sequence of measures $\mathbb{P}_n^F$ over the full data $(\bar{A}, C, T_{\mathcal{A}}, \bar{L}_{\mathcal{A}})$ satisfying Assumptions 1, 2, 3, 4, whose inductions on the observed data converges to $\mathbb{P}$ in the total variation norm.*

Technically, we have not achieved full nonparametric paradigm. However, we deem that the regularity condition "the path space is piece-wise continuous processes" is general enough for practical considerations. For example, both multivariate counting processes and continuous processes like Brownian process satisfy this regularity condition. It is noteworthy that achieving this "almost nonparametric" nature is the best one can hope for. This realization was confirmed in (Gill et al., 1997, Section 9) for "coarsening at random" assumption, even though our framework exhibits certain distinctions.

## 4 EXPERIMENT RESULT

In this section, we employ Monte Carlo simulations to empirically assess how the identification works. We decide to evaluate the performance of the g-computation formula only, for two reasons:

1. The g-computation formula is the only one that can be easily approximated through raw simulated data, whereas inverse probability weighting (and hence the doubly robust formula) cannot be directly approximated without estimation or computation. In fact, in causal inference with longitudinal data, the true causal effects are often not analytically computable. Instead, they are approximated numerically using methods like the g-computation formula through sampling like we outline below, with very large sample sizes, a standard practice for benchmarking estimator performance;

2. On the population level, the values of the three formulas are all the same, equaling (1). Therefore, approximating g-computation formula is sufficient for our purposes.

To that end, we need to go through 4 steps:

1. Come up with a reasonable data generating process;

2. Compute the parameter of interest (1) (or equivalently, the left-hand side of g-computation formula in Theorem 1) according to this data generating process;

3. Simulate according to this data generating process;

4. Approximate the right-hand side of g-computation formula in Theorem 1 using the simulated data.

**Step 1**: To sharp the focus and ease the computation, we consider a simple setting where there is no mortality or censoring ($T = C \equiv \infty$), or other measured confounding process, except for the outcome process itself. A more complicated scenario including mortality and censoring, and other confounding process, is considered in Section D in the appendix. We take glucose levels as the outcome and insulin levels as the treatment. Both glucose and insulin levels exhibit smooth, continuous changes over time. Gaussian processes are particularly well-suited for modeling such smooth and continuous temporal processes. For $t \in [0, 1]$, consider a potential outcome process $Y_{\bar{a}}(t)$ capturing potential logarithm of glucose levels, following a Gaussian process with mean process as

$$\mathbb{E}(Y_{\bar{a}}(t)) = -a(t), \tag{24}$$

and covariance process as

$$\text{Cov}[Y_{\bar{a}}(t), Y_{\bar{a}}(s)] = e^{-3|t-s|}, \ \forall t, s \in [0, 1]. \tag{25}$$

This ensures the joint dependence among $Y_{\bar{a}}(t)$ and negative treatment effect of logarithm of insulin level $\bar{a}$. For instance, $Y_{\bar{a}}(t)$ can be log of blood glucose level. Define $\nu(T_{\bar{a}}, \bar{Y}_{\bar{a}})$ as the integral of $\bar{Y}_{\bar{a}}$ over time $t \in [0, 1]$, that is,

$$\nu(T_{\bar{a}}, \bar{Y}_{\bar{a}}) = \int_0^1 Y_{\bar{a}}(t)dt. \tag{26}$$

Suppose the targeted treatment regime $\mathbb{G}$ is a Gaussian measure with mean process $t - 0.5$ and jointly independent normal variables at any time points. That is, the intervened $A$ follows a Gaussian process with a mean process

$$\mathbb{E}(A(t)) = t - 0.5, \tag{27}$$

and covariance process

$$\text{Cov}[A(t), A(s)] = e^{-3|t-s|}, \ \forall t, s \in [0, 1], \tag{28}$$

representing an increase of insulin level, possibly due to some insulin intake.

**Step** : Then we can show that (1) (or equivalently, the left-hand side of g-computation formula in Theorem 1) equals zero, that is,

$$\int \mathbb{E}(\nu(T_{\bar{a}}, \bar{Y}_{\bar{a}}))\mathbb{G}(\mathrm{d}\bar{a}) = \int \mathbb{E}\left[\int_0^1 Y_{\bar{a}}(t)dt\right]\mathbb{G}(\mathrm{d}\bar{a}) = \int \int_0^1 \mathbb{E}(Y_{\bar{a}}(t))dt\mathbb{G}(\bar{a}) \qquad (29)$$

$$= \int \left[\int_0^1 a(t)dt\right]\mathbb{G}(\mathrm{d}\bar{a}) = \int_0^1 \int a(t)\mathbb{G}(\bar{a}(t))dt = \int_0^1 (t-0.5)dt = 0. \qquad (30)$$

**Step 3**: In practice, we observe a stochastic process at finite points. We consider evenly splitting $t \in [0,1]$ into a grid of size $K+1$: $\Delta_K[0,1] = \{t_0 = 0, t_1 = 1/K, \cdots, t_{K-1} = (K-1)/K, t_K = 1\}$, and for $1 \leq i \leq n$, according to $\mathbb{G}$ specified in Step 1, we simulate i.i.d. samples $A_i(t)$ according to $\mathbb{G}$ specified in Step 1 at $\Delta_K[0,1]$ as

$$\begin{pmatrix} A_i(t_0) \\ A_i(t_1) \\ \cdots \\ A_i(t_{K-1}) \\ A_i(t_K) \end{pmatrix} \sim \mathcal{N}\left[\begin{pmatrix} t_0 - 0.5 \\ t_1 - 0.5 \\ \cdots \\ t_{K-1} - 0.5 \\ t_K - 0.5 \end{pmatrix}, \begin{pmatrix} 1 & e^{-3|t_1-t_0|} & \cdots & e^{-3|t_{K-1}-t_0|} & e^{-3|t_K-t_0|} \\ e^{-3|t_1-t_0|} & 1 & \cdots & e^{-3|t_{K-1}-t_1|} & e^{-3|t_K-t_1|} \\ \cdots & \cdots & \cdots & \cdots & \cdots \\ e^{-3|t_{K-1}-t_0|} & e^{-3|t_{K-1}-t_1|} & \cdots & 1 & e^{-3|t_K-t_{K-1}|} \\ e^{-3|t_K-t_0|} & e^{-3|t_K-t_1|} & \cdots & e^{-3|t_K-t_{K-1}|} & 1 \end{pmatrix}\right].$$

By according to the distribution of $Y_{\bar{a}}(t)$ specified in Step 1 and consistency, we generate $Y_i(t)$ at $\Delta_K[0,1]$ as

$$\begin{pmatrix} Y_i(t_0) \\ Y_i(t_1) \\ \cdots \\ Y_i(t_{K-1}) \\ Y_i(t_K) \end{pmatrix} \sim \mathcal{N}\left[\begin{pmatrix} A_i(t_0) \\ A_i(t_1) \\ \cdots \\ A_i(t_{K-1}) \\ A_i(t_K) \end{pmatrix}, \begin{pmatrix} 1 & e^{-|t_1-t_0|} & \cdots & e^{-|t_{K-1}-t_0|} & e^{-|t_K-t_0|} \\ e^{-|t_1-t_0|} & 1 & \cdots & e^{-|t_{K-1}-t_1|} & e^{-|t_K-t_1|} \\ \cdots & \cdots & \cdots & \cdots & \cdots \\ e^{-|t_{K-1}-t_0|} & e^{-|t_{K-1}-t_1|} & \cdots & 1 & e^{-|t_K-t_{K-1}|} \\ e^{-|t_K-t_0|} & e^{-|t_K-t_1|} & \cdots & e^{-|t_K-t_{K-1}|} & 1 \end{pmatrix}\right].$$

**Step 4**: The integral of $Y_i(t_k)$ over $[0,1]$ is $\sum_{k=0}^K Y_i(t_k)/(K+1)$. The approximate of the right-hand side of g-computation formula is $\sum_{i=1}^n \sum_{k=0}^K Y_i(t_k)/(K+1)/n$.

We vary the grid sizes ($K = 10, 50, 250$) to examine how a denser grid improves the approximation. This approach simulates the scenario where the mesh $|\Delta_K[0,1]|$ is shrunk to zero. Additionally, we vary the sample sizes ($n = 100, 500, 2500$) to explore how larger samples enhance the approximation, leveraging the law of large numbers to better approximate the right-hand side of the g-computation formula. We repeat the process $R = 10{,}000$ times. The resulting 10,000 approximations of $\sum_{i=1}^n \sum_{k=0}^K Y_i(t_k)/(K+1)/n$ are presented in boxplots in Figure 1, where we append biases.

The simulation results demonstrate that the g-computation formula can adequately approximate (1) even with moderate sample and grid sizes. Increasing the sample size while keeping the grid size fixed enhances the accuracy and reduces the variance of the approximation. In contrast, increasing the grid size while keeping the sample size fixed does not consistently improve accuracy or reduce variance. However, simultaneously increasing both the sample and grid sizes significantly improves accuracy and reduces variance in the approximation.

## 5 CONCLUSION

In this work, we proposed on a novel theoretical framework for causal inference under functional longitudinal studies. We introduced three methodological paradigms for causal identification: the g-computation formula, inverse probability weighting formula, and doubly robust formula. This framework, noted for nonparametric foundation, substantiates and expands upon the estimand-based causal framework introduced by Ying (2024a). It incorporates considerations for time-varying outcomes and addresses complexities such as death and right censoring, marking a significant advancement in the analysis of functional longitudinal data and enhancing the toolkit for causal inference in this area.

Our focus is on the underlying curve data (Wang et al., 2016). At the population level, our framework abstracts away the sparsity or regularity of sample-level observations. In future work, we plan to extend our framework to sample-level data, where factors such as sparsity or irregularity could

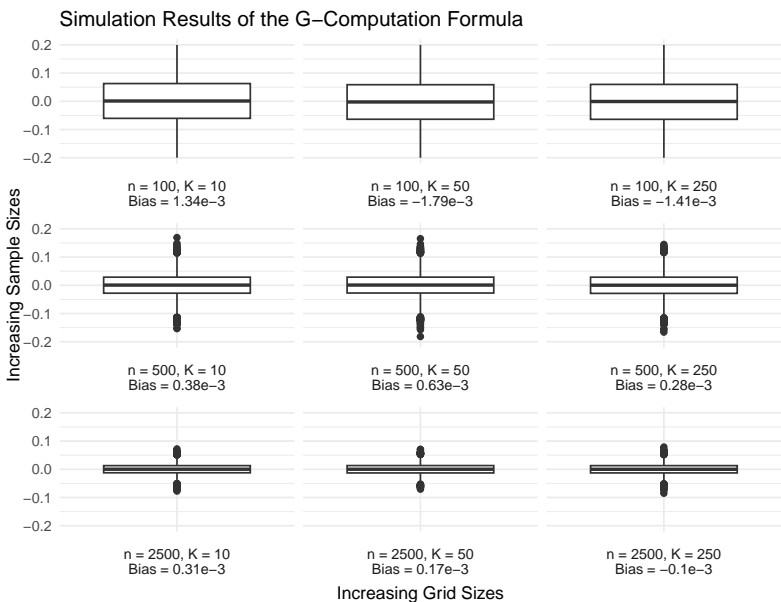

Figure 1: Simulation Results of using g-computation formula by varying grid sizes in $K = 10, 50, 250$ and sample sizes in $n = 100, 500, 2500$, for $R = 10000$ repeats. We plot boxplots and give biases.

influence the consistency of estimators. For example, investigating how the number of observed time points $p_n$ scales with the sample size $n$ in densely observed data could provide valuable insights.

There are significant theoretical and methodological opportunities, given the limited investigation on functional longitudinal data, for the machine learning, functional data analysis and causal inference communities. To list a few, first, adapting our framework to accommodate scenarios where Assumption 1 may not hold, including contexts involving time-dependent instrumental variables and time-dependent proxies (Ying et al., 2023), warrants rigorous exploration. Following the same spirit, dependent censoring can be considered, for instance, generalizing proxy method like Ying (2024c). Second, the positivity Assumption 4 in longitudinal studies faces practical challenges due to the potential scarcity of subjects adhering to specific treatment regimes within observed populations. One might consider using semiparametric models such as marginal structural models (Robins, 1998; Røysland, 2011) and structural nested models (Robins, 1999; Lok, 2008). Other solutions include dynamic treatment regimes (Fitzmaurice et al., 2008; Young et al., 2011; Rytgaard et al., 2022) and incremental interventions (Kennedy, 2017). Third, establishing the efficiency bound for our quantity of interest by leveraging semiparametric theory, represents an engaging challenge. Fourth, partial identification using discrete-time observations is a promising direction. Finally, developing a comprehensive estimation framework remains of ultimate interest.

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

## A  FORMAL FORMULAS

For any partition $\Delta_K[0, \infty]$, we have the following decomposition

$$\mathbb{P}(\mathrm{d}x\mathrm{d}\delta\mathrm{d}\bar{a}\mathrm{d}\bar{l}) \tag{31}$$

$$= \prod_{j=0}^{K-1} \mathbb{P}[T \leq t_{j+1}|\bar{l}(t_{j+1}), \bar{a}(t_{j+1}), \mathbb{1}(t_j < x \leq t_{j+1}, \delta = 0), \mathscr{F}_{t_j}]^{\mathbb{1}(t_j < x \leq t_{j+1}, \delta=1)} \tag{32}$$

$$\mathbb{P}[T > t_{j+1}|\bar{l}(t_{j+1}), \bar{a}(t_{j+1}), \mathbb{1}(t_j < x \leq t_{j+1}, \delta = 0), \mathscr{F}_{t_j}]^{\mathbb{1}(x \leq t_{j+1}, \delta=0)} \tag{33}$$

$$\mathbb{P}[C \leq t_{j+1}|\bar{l}(t_{j+1}), \bar{a}(t_{j+1}), \mathscr{F}_{t_j}]^{\mathbb{1}(t_j < x \leq t_{j+1}, \delta=0)} \tag{34}$$

$$\mathbb{P}[C > t_{j+1}|\bar{l}(t_{j+1}), \bar{a}(t_{j+1}), \mathscr{F}_{t_j}]^{\mathbb{1}(x \leq t_{j+1}, \delta=1)} \tag{35}$$

$$\mathbb{P}[\mathrm{d}\bar{l}(t_{j+1})|\bar{a}(t_{j+1}), \mathscr{F}_{t_j}] \tag{36}$$

$$\mathbb{P}[\mathrm{d}\bar{a}(t_{j+1})|\mathscr{F}_{t_j}]. \tag{37}$$

We intervene treatment distribution at each time $t_j$ to approximate the pseudo-population where $\bar{A}$ were to follow $\mathbb{G}$ as:

$$\mathbb{P}_{\Delta_K[0,\infty],\mathbb{G}}(\mathrm{d}x\mathrm{d}\delta\mathrm{d}\bar{a}\mathrm{d}\bar{l}) \tag{38}$$

$$= \prod_{j=0}^{K-1} \mathbb{P}[T \leq t_{j+1}|\bar{l}(t_{j+1}), \bar{a}(t_{j+1}), \mathscr{F}_{t_j}]^{\mathbb{1}(t_j < x \leq t_{j+1}, \delta=1)} \tag{39}$$

$$[1 - \mathbb{1}(x \leq t_{j+1}, \delta = 0)] \tag{40}$$

$$\mathbb{P}[\mathrm{d}\bar{l}(t_{j+1})|\bar{a}(t_{j+1}), \mathscr{F}_{t_j}] \tag{41}$$

$$\mathbb{G}[\mathrm{d}\bar{a}(t_{j+1})|\bar{a}(t_j)]. \tag{42}$$

## B  A REVIEW OF IDENTIFICATION FORMULAS FOR REGULAR LONGITUDINAL STUDIES

The following is a review of existing methods and rewritten by our language. Consider a longitudinal study with data collected at fixed times $t = 0, \ldots, K$. To align with the notation used in this paper, let $\tau = K$. Here, $A(t)$ and $L(t)$ can only change at discrete time points $t = k$. The associated filtrations are defined as $\mathscr{F}_k = \sigma(\bar{A}(k), \bar{L}(k))$, $\mathscr{F}_{k-} = \sigma(\bar{A}(k-1), \bar{L}(k-1))$, and $\mathscr{G}_k = \sigma(\bar{A}(k), \bar{L}(k-1))$. For a finite set of random variables, the measure on the path space induced by $\mathbb{P}$ corresponds to a multivariate distribution. Denote the observed data density or probability mass function as $p(\bar{a}, \bar{l})$, and use $p(\cdot|\cdot)$ for conditional densities or probabilities.

The observed data likelihood can be uniquely factorized based on the temporal sequence of events:

$$p(\bar{a}, \bar{l}) = \prod_{j=0}^{K} \left[ p[l(j)|\bar{a}(j), \bar{l}(j-1)]p[a(j)|\bar{a}(j-1), \bar{l}(j-1)] \right], \tag{43}$$

where $\bar{l}(-1) = \bar{a}(-1) = \emptyset$ for notational convenience.

To identify (1), the following positivity assumption is imposed:

$$\mathrm{g}(\bar{a}) \ll \prod_{j=0}^{K} p\{a(j)|\bar{a}(j-1), \bar{L}(j-1)\}, \tag{44}$$

almost surely over $\bar{L}$. Alternatively, a stricter version requires:

$$p\{a(j)|\bar{a}(j-1), \bar{L}(j-1)\} > 0, \tag{45}$$

almost surely over $\bar{L}(j-1)$ for any $\bar{a}$. For example, in the case of a binary treatment, if $\mathrm{g}(\bar{A}) = \mathbb{1}(\bar{A} = \bar{0})$ (i.e., "always under control"), the positivity assumption ensures that for any patient history $\bar{L}$, the probability of remaining under control is nonzero.

Given the positivity assumption, one can substitute $p[a(j)|\bar{a}(j-1), \bar{l}(j-1)]$ in (43) with $g\{a(j)|\bar{a}(j-1)\}$, leading to the target distribution:

$$p_{\mathbb{G}}(\bar{a}, \bar{l}) = \prod_{j=0}^{K} \left\{ p[l(j)|\bar{a}(j), \bar{l}(j-1)] p_{\mathbb{G}}[a(j)|\bar{a}(j-1)] \right\}. \tag{46}$$

This distribution enables the identification of $\mathbb{E}_{\mathbb{G}}(\nu(L(K)))$, where $\mathbb{E}_{\mathbb{G}}$ represents the expectation under $p_{\mathbb{G}}$. Additional assumptions are required to interpret $\mathbb{E}_{\mathbb{G}}(\nu(L(K)))$ causally:

$$\int_{\mathcal{A}} \mathbb{E}(\nu(L_{\bar{a}}(K))) g(\bar{a}) d\bar{a} = \mathbb{E}_{\mathbb{G}}(\nu(L(K))). \tag{47}$$

The first assumption is consistency:

$$L(K) = L_{\bar{A}}(K), \tag{48}$$

almost surely. This states that the observed outcome matches the potential outcome under the received treatment regime $\bar{A} = \bar{a}$.

The second is the sequential randomization assumption (SRA), also referred to as "no unmeasured confounders." It posits that the treatment $A(k)$ at time $k$ is conditionally independent of the potential outcome $L_{\bar{a}}(K)$ given past treatment and covariate history:

$$A(k) \perp L_{\bar{a}}(K) \mid \bar{A}(k-1), \bar{L}(k-1), \tag{49}$$

for any $\bar{a} \in \mathcal{A}$ and $k = 0, \dots, K$. This ensures exchangeability across treatment groups within strata defined by observed covariates. Note that alternative frameworks relax this assumption (Tchetgen Tchetgen et al., 2018; 2020; Ying et al., 2023), but they are beyond the scope of this discussion.

The g-computation formula (Greenland & Robins, 1986; Robins, 2000) and inverse probability weighting (IPW) (Hernán et al., 2000; 2002) are two commonly used approaches for identification. The g-computation formula identifies (47) through iterative conditional expectations:

$$\int_{\mathcal{A}} \mathbb{E}(\nu(L_{\bar{a}}(K))) g(\bar{a}) d\bar{a} = \int \nu(l(K)) \prod_{j=0}^{K} \left\{ p[l(j)|\bar{a}(j), \bar{l}(j-1)] g\{a(j)|\bar{a}(j-1)\} dl(j) da(j) \right\}. \tag{50}$$

Alternatively, the IPW formula uses pseudoweights derived from the Radon-Nikodym derivative:

$$\int_{\mathcal{A}} \mathbb{E}(\nu(L_{\bar{a}}(K))) g(\bar{a}) d\bar{a} = \mathbb{E} \left\{ \frac{g(\bar{A}) \nu(L(K))}{\prod_{j=0}^{K} p[A(j)|\bar{A}(j-1), \bar{L}(j-1)]} \right\}. \tag{51}$$

Lastly, the doubly robust (DR) formula combines both approaches (Bang & Robins, 2005; Van der Laan & Robins, 2003):

$$\int_{\mathcal{A}} \mathbb{E}(\nu(L_{\bar{a}}(K))) g(\bar{a}) d\bar{a} \tag{52}$$

$$= \mathbb{E} \left( Q_{\mathbb{G}}(K) \nu(L(K)) - \sum_{k=0}^{K} \left\{ Q_{\mathbb{G}}(k) H_{\mathbb{G}}(k) - Q_{\mathbb{G}}(k-1) \int H_{\mathbb{G}}(k) g[a(k)|\bar{A}(k-1)] da(k) \right\} \right). \tag{53}$$

For additional details, see Robins (1986; 1997); Hernán & Robins (2020).

## C  PROOFS

### C.1  PROOF OF PROPOSITION 1

We temporarily define $\mathscr{F}_{/C,t} = \sigma(\{L(s), A(s), \mathbb{1}(T \leq s) : \forall s \leq t\})$ as the censoring free filtration and $\mathscr{F}_{\bar{a},t} = \sigma(\{L_{\bar{a}}(s), \mathbb{1}(T_{\bar{a}} \leq s) : \forall s \leq t\})$ as the counterfactual filtration:

$$
\left\| \mathbb{P}_{\Delta_K[0,\infty],\mathbb{G}}(\mathrm{d}x\mathrm{d}\delta\mathrm{d}\bar{a}\mathrm{d}\bar{l}) - \mathbb{P}(\mathrm{d}x_{\bar{a}}\mathrm{d}l_{\bar{a}})\mathbb{G}(\mathrm{d}\bar{a})\delta_{\bar{a}} \right\|_{\mathrm{TV}}
$$

$$
= \left\| \prod_{j=0}^{K-1} \mathbb{G}(\mathrm{d}\bar{a}(t_{j+1})|\bar{a}(t_j))[1 - \mathbb{1}(x \leq t_{j+1}, \delta = 0)] \, \mathbb{P}(T \leq t_{j+1}|\bar{a}(t_{j+1}), \mathscr{F}_{t_j})^{\mathbb{1}(t_j < x \leq t_{j+1})} \right.
$$

$$
\mathbb{P}(T > t_{j+1}|\bar{a}(t_{j+1}), \mathscr{F}_{t_j})^{1-\mathbb{1}(t_j < x \leq t_{j+1})} \, \mathbb{P}(\mathrm{d}\bar{l}(t_{j+1})|\bar{a}(t_j), \mathscr{F}_{t_j})
$$

$$
\left. - \mathbb{P}(\mathrm{d}x_{\bar{a}}\mathrm{d}l_{\bar{a}})\mathbb{G}(\mathrm{d}\bar{a})\delta_{\bar{a}} \right\|_{\mathrm{TV}}
$$

$$
\leq \left\| \prod_{j=0}^{K-1} \mathbb{G}(\mathrm{d}\bar{a}(t_{j+1})|\bar{a}(t_j))[1 - \mathbb{1}(x \leq t_{j+1}, \delta = 0)] \right.
$$

$$
\mathbb{P}(T \leq t_{j+1}|\mathscr{F}_{/C,t_j})^{\mathbb{1}(t_j < x \leq t_{j+1})} \, \mathbb{P}(T > t_{j+1}|\mathscr{F}_{/C,t_j})^{1-\mathbb{1}(t_j < x \leq t_{j+1})} \, \mathbb{P}(\mathrm{d}\bar{l}(t_{j+1})|\mathscr{F}_{/C,t_j})
$$

$$
\left. - \mathbb{P}(\mathrm{d}x_{\bar{a}}\mathrm{d}l_{\bar{a}})\mathbb{G}(\mathrm{d}\bar{a})\delta_{\bar{a}} \right\|_{\mathrm{TV}} + o(1)
$$

$$
= \left\| \prod_{j=0}^{K-1} \mathbb{G}(\mathrm{d}\bar{a}(t_{j+1})|\bar{a}(t_j))\{1 - \mathbb{1}(x_{\bar{a}} \leq t_{j+1}, \delta_{\bar{a}} = 0)\} \, \mathbb{P}(T_{\bar{a}} \leq t_{j+1}|\bar{a}(t_{j+1}), \mathscr{F}_{\bar{a},t_j})^{\mathbb{1}(t_j < x_{\bar{a}} \leq t_{j+1}, \delta_{\bar{a}}=1)} \right.
$$

$$
\mathbb{P}(T_{\bar{a}} > t_{j+1}|\bar{a}(t_{j+1}), \mathscr{F}_{\bar{a},t_j})^{1-\mathbb{1}(t_j < x_{\bar{a}} \leq t_{j+1}, \delta_{\bar{a}}=1)} \, \mathbb{P}(\mathrm{d}\bar{l}_{\bar{a}}(t_{j+1})|\bar{a}(t_{j+1}), \mathscr{F}_{\bar{a},t_j})
$$

$$
\left. - \mathbb{P}(\mathrm{d}x_{\bar{a}}\mathrm{d}l_{\bar{a}})\mathbb{G}(\mathrm{d}\bar{a})\delta_{\bar{a}} \right\|_{\mathrm{TV}}
$$

$$
= \left\| \prod_{j=0}^{K-1} \mathbb{G}(\mathrm{d}\bar{a}(t_{j+1})|\bar{a}(t_j))[1 - \mathbb{1}(x_{\bar{a}} \leq t_{j+1}, \delta_{\bar{a}} = 0)] \right.
$$

$$
\{\mathbb{P}(T_{\bar{a}} \leq t_{j+1}|C > t_{j+1}, \bar{a}(t_{j+1}), \mathscr{F}_{\bar{a},t_j})^{\mathbb{1}(t_j < x_{\bar{a}} \leq t_{j+1}, \delta_{\bar{a}}=1)}
$$

$$
\mathbb{P}(T_{\bar{a}} > t_{j+1}|C > t_{j+1}, \bar{a}(t_{j+1}), \mathscr{F}_{\bar{a},t_j})^{1-\mathbb{1}(t_j < x_{\bar{a}} \leq t_{j+1}, \delta_{\bar{a}}=1)} \, \mathbb{P}(\mathrm{d}\bar{l}_{\bar{a}}(t_{j+1})|C > t_j, \bar{a}(t_{j+1}), \mathscr{F}_{\bar{a},t_j})
$$

$$
\left. - \mathbb{P}(T_{\bar{a}} \leq t_{j+1}|\mathscr{F}_{\bar{a},t_j})^{\mathbb{1}(t_j < x_{\bar{a}} \leq t_{j+1})} \, \mathbb{P}(T_{\bar{a}} > t_{j+1}|\mathscr{F}_{\bar{a},t_j})^{1-\mathbb{1}(t_j < x_{\bar{a}} \leq t_{j+1})} \, \mathbb{P}(\mathrm{d}\bar{l}_{\bar{a}}(t_{j+1})|\mathscr{F}_{\bar{a},t_j})\} \right\|_{\mathrm{TV}} \to 0.
$$

### C.2  PROOF OF THEOREM 1

Since $\mathbb{P}_{\mathbb{G}}(\mathrm{d}x\mathrm{d}\delta\mathrm{d}\bar{a}\mathrm{d}\bar{l})$ is a limit of measures in total variation norm of

$$
\mathbb{P}_{\Delta_K[0,\infty],\mathbb{G}}(\mathrm{d}x\mathrm{d}\delta\mathrm{d}\bar{a}\mathrm{d}\bar{l}), \tag{54}
$$

whenever $|\Delta_K[t,\infty]| \to 0$, we have

$$
\int f(x, \delta, \bar{a}, \bar{l}) \, \mathbb{P}_{\Delta_K[0,\infty],\mathbb{G}}(\mathrm{d}x\mathrm{d}\delta\mathrm{d}\bar{a}\mathrm{d}\bar{l}) \to \mathbb{E}_{\mathbb{G}}[f(X, \Delta, \bar{A}, \bar{L})], \tag{55}
$$

for any bounded functions $f(x, \delta, \bar{a}, \bar{l})$. Therefore, we have

$$
\left| H_{\mathbb{G}}(0-) - \int \mathbb{E}(\nu(T_{\bar{a}}, Y_{\bar{a}})) \mathbb{G}(\mathrm{d}\bar{a}) \right|
$$

$$
\leq \left| \int \nu(x, \bar{y}) \, \mathbb{P}(\mathrm{d}x\mathrm{d}\delta\mathrm{d}\bar{a}\mathrm{d}\bar{l}) - \int \nu(x, \bar{y}) \, \mathbb{P}_{\Delta_K[0,\infty],\mathbb{G}}(\mathrm{d}x\mathrm{d}\delta\mathrm{d}\bar{a}\mathrm{d}\bar{l}) \right|
$$

$$
+ \left| \int \nu(x, \bar{y}) \, \mathbb{P}_{\Delta_K[0,\infty],\mathbb{G}}(\mathrm{d}x\mathrm{d}\delta\mathrm{d}\bar{a}\mathrm{d}\bar{l}) - \int \mathbb{E}(\nu(T_{\bar{a}}, Y_{\bar{a}})) \mathbb{G}(\mathrm{d}\bar{a}) \right|
$$

$$
= \left| \int \nu(x, \bar{y}) \, \mathbb{P}_{\Delta_K[0,\infty],\mathbb{G}}(\mathrm{d}x\mathrm{d}\delta\mathrm{d}\bar{a}\mathrm{d}\bar{l}) - \int \mathbb{E}(\nu(T_{\bar{a}}, Y_{\bar{a}})) \mathbb{G}(\mathrm{d}\bar{a}) \right| + o(1)
$$

$$
= \left| \int \nu(x, \bar{y}) \prod_{j=0}^{K-1} \mathbb{P}(T \leq t_{j+1} | C > t_{j+1}, \bar{a}(t_{j+1}), \mathscr{F}_{t_j})^{\mathbb{1}(t_j < x \leq t_{j+1}, \delta=1)} \right.
$$

$$
\mathbb{P}(T > t_{j+1} | C > t_{j+1}, \bar{a}(t_{j+1}), \mathscr{F}_{t_j})^{1-\mathbb{1}(t_j < x \leq t_{j+1}, \delta=1)} \, \mathbb{P}(\mathrm{d}\bar{l}(t_{j+1}) | C > t_{j+1}, \bar{a}(t_{j+1}), \mathscr{F}_{t_j})
$$

$$
\left. [1 - \mathbb{1}(x \leq t_{j+1}, \delta = 0)] \mathbb{G}(\mathrm{d}\bar{a}(t_{j+1}) | \bar{a}(t_j)) - \int \mathbb{E}(\nu(T_{\bar{a}}, Y_{\bar{a}})) \mathbb{G}(\mathrm{d}\bar{a}) \right| + o(1),
$$

where $o(1)$ converges to zero when $\Delta_K[0, \infty] \to 0$. By Assumptions 1, 2, 3, and 4, the above term is less than or equal to

$$
\left| \int \nu(x, \bar{y}) \, \mathbb{P}(T \leq t_K | \bar{a}(t_K), \mathscr{F}_{t_{K-1}})^{\mathbb{1}(t_{K-1} < x \leq t_K, \delta=1)} \right.
$$

$$
\mathbb{P}(T > t_K | \bar{a}(t_K), \mathscr{F}_{t_{K-1}})^{1-\mathbb{1}(t_{K-1} < x \leq t_K, \delta=1)} \, \mathbb{P}(\mathrm{d}\bar{l}(t_K) | \bar{a}(t_K), \mathscr{F}_{t_{K-1}})
$$

$$
[1 - \mathbb{1}(x \leq t_K, \delta = 0)] \mathbb{G}(\mathrm{d}\bar{a}(t_K) | \bar{a}(t_{K-1})) \prod_{j=0}^{K-1} \mathbb{P}(T \leq t_{j+1} | \bar{a}(t_{j+1}), \mathscr{F}_{/C,t_j})^{\mathbb{1}(t_j < x \leq t_{j+1}, \delta=1)}
$$

$$
\mathbb{P}(T > t_{j+1} | \bar{a}(t_{j+1}), \mathscr{F}_{/C,t_j})^{1-\mathbb{1}(t_j < x \leq t_{j+1}, \delta=1)} \, \mathbb{P}(\mathrm{d}\bar{l}(t_{j+1}) | \bar{a}(t_{j+1}), \mathscr{F}_{/C,t_j})
$$

$$
\left. [1 - \mathbb{1}(x \leq t_{j+1}, \delta = 0)] \mathbb{G}(\mathrm{d}\bar{a}(t_{j+1}) | \bar{a}(t_j)) - \int \mathbb{E}(\nu(T_{\bar{a}}, Y_{\bar{a}})) \mathbb{G}(\mathrm{d}\bar{a}) \right| + o(1),
$$

which by Assumptions 1, 2, 3, and 4, equals

$$
\left| \int \nu(x, \bar{y}) \, \mathbb{P}(T_{\bar{a}} \leq t_K | \bar{a}(t_K), \mathscr{F}_{t_{K-1}})^{\mathbb{1}(t_{K-1} < x_{\bar{a}} \leq t_K, \delta_{\bar{a}}=1)} \right.
$$

$$
\mathbb{P}(T_{\bar{a}} > t_K | \bar{a}(t_K), \mathscr{F}_{t_{K-1}})^{1-\mathbb{1}(t_{K-1} < x_{\bar{a}} \leq t_K, \delta_{\bar{a}}=1)} \, \mathbb{P}(\mathrm{d}\bar{l}_{\bar{a}}(t_K) | \bar{a}(t_K), \mathscr{F}_{t_{K-1}})
$$

$$
[1 - \mathbb{1}(x_{\bar{a}} \leq t_K, \delta_{\bar{a}} = 0)] \mathbb{G}(\mathrm{d}\bar{a}(t_K) | \bar{a}(t_{K-1})) \prod_{j=0}^{K-1} \mathbb{P}(T \leq t_{j+1} | \bar{a}(t_{j+1}), \mathscr{F}_{/C,t_j})^{\mathbb{1}(t_j < x \leq t_{j+1}, \delta=1)}
$$

$$
\mathbb{P}(T > t_{j+1} | \bar{a}(t_{j+1}), \mathscr{F}_{/C,t_j})^{1-\mathbb{1}(t_j < x \leq t_{j+1}, \delta=1)} \, \mathbb{P}(\mathrm{d}\bar{l}(t_{j+1}) | \bar{a}(t_{j+1}), \mathscr{F}_{/C,t_j})
$$

$$
\left. [1 - \mathbb{1}(x \leq t_{j+1}, \delta = 0)] \mathbb{G}(\mathrm{d}\bar{a}(t_{j+1}) | \bar{a}(t_j)) - \int \mathbb{E}(\nu(T_{\bar{a}}, Y_{\bar{a}})) \mathbb{G}(\mathrm{d}\bar{a}) \right| + o(1),
$$

which by Assumptions 1, 2, 3, and 4, is less than or equal to

$$
\left| \int \nu(x,\bar{y})\, \mathbb{P}(T_{\bar{a}} \leq t_K | \mathscr{F}_{t_{K-1}})^{\mathbb{1}(t_{K-1} < x_{\bar{a}} \leq t_K, \delta_{\bar{a}}=1)} \right.
$$

$$
\mathbb{P}(T_{\bar{a}} > t_K | \mathscr{F}_{t_{K-1}})^{1-\mathbb{1}(t_{K-1} < x_{\bar{a}} \leq t_K, \delta_{\bar{a}}=1)}\, \mathbb{P}(\mathrm{d}\bar{l}_{\bar{a}}(t_K)|\mathscr{F}_{t_{K-1}})
$$

$$
[1 - \mathbb{1}(x_{\bar{a}} \leq t_K, \delta_{\bar{a}} = 0)]\mathbb{G}(\mathrm{d}\bar{a}(t_K)|\bar{a}(t_{K-1}))\prod_{j=0}^{K-1}\mathbb{P}(T \leq t_{j+1}|\bar{a}(t_{j+1}), \mathscr{F}_{/C,t_j})^{\mathbb{1}(t_j < x \leq t_{j+1}, \delta=1)}
$$

$$
\mathbb{P}(T > t_{j+1}|\bar{a}(t_{j+1}), \mathscr{F}_{/C,t_j})^{1-\mathbb{1}(t_j < x \leq t_{j+1}, \delta=1)}\, \mathbb{P}(\mathrm{d}\bar{l}(t_{j+1})|\bar{a}(t_{j+1}), \mathscr{F}_{/C,t_j})
$$

$$
\left. [1 - \mathbb{1}(x \leq t_{j+1}, \delta = 0)]\mathbb{G}(\mathrm{d}\bar{a}(t_{j+1})|\bar{a}(t_j)) - \int \mathbb{E}(\nu(T_{\bar{a}}, Y_{\bar{a}}))\mathbb{G}(\mathrm{d}\bar{a}) \right| + o(1).
$$

By iterating the above process for $0 \leq j \leq K-2$, we arrive the conclusion.

## C.3 PROOF OF THEOREM 2

The proof is immediate by noting that

$$
\mathbb{E}\left[Q_{\mathbb{G}}(\infty)\nu(X,\bar{Y})\right] = \mathbb{E}_{\mathbb{G}}[\nu(X,\bar{Y})] = \int_{\mathcal{A}} \mathbb{E}(\nu(T_{\bar{a}}, \bar{Y}_{\bar{a}}))\mathbb{G}(\mathrm{d}\bar{a}), \tag{56}
$$

by Theorem 1.

## C.4 PROOF OF THEOREM 3

We first prove the theorem when $H = H_{\mathbb{G}}$. Indeed, as the limit and expectation can interchange, we can show that

$$
\left| \mathbb{E}\left[ \Xi_{\mathrm{out},\Delta_K[0,\infty]}(H_{\mathbb{G}}, Q) \right] \right|
$$

$$
= \left| \mathbb{E}\left\{ Q(t_K)[\nu(X,\bar{Y}) - H_{\mathbb{G}}(t_K)] \right\} \right.
$$

$$
\left. + \sum_{j=1}^{K-1} \mathbb{E}\left( Q(t_j) \right\} \int H_{\mathbb{G}}(t_{j+1})\mathbb{G}(\mathrm{d}\bar{a}(t_{j+1})|\bar{A}(t_j)) - H_{\mathbb{G}}(t_j) \right\} ) \right|
$$

$$
= \left| 0 + \sum_{j=1}^{K-1} \mathbb{E}\left( Q(t_j)\left\{ \int H_{\mathbb{G}}(t_{j+1})\mathbb{G}(\mathrm{d}\bar{a}(t_{j+1})|\bar{A}(t_j)) - H_{\mathbb{G}}(t_j) \right\} \right) \right|
$$

$$
\leq \sum_{j=1}^{K-1} \left| \mathbb{E}\left( Q(t_j)\left\{ \int H_{\mathbb{G}}(t_{j+1})\mathbb{G}(\mathrm{d}\bar{a}(t_{j+1})|\bar{A}(t_j)) - H_{\mathbb{G}}(t_j) \right\} \right) \right|
$$

$$
\leq \sum_{j=1}^{K-1} \left| \mathbb{E}\left( Q(t_j)\left\{ \int H_{\mathbb{G}}(t_{j+1})\mathbb{G}(\mathrm{d}\bar{a}(t_{j+1})|\bar{A}(t_j)) - \mathbb{E}_{\mathbb{G}}\left[ H_{\mathbb{G}}(t_{j+1})|\mathscr{G}_{t_j} \right] \right\} \right) \right|
$$

$$
\leq \sum_{j=0}^{K} \kappa \|H_{\mathbb{G}}(t_j)Q(t_j)\|_1 (t_{j+1} - t_j)^{\alpha}
$$

$$
\leq \kappa \sup_t \|H_{\mathbb{G}}(t)Q(t)\|_1 \sum_{j=1}^{K-1} (t_{j+1} - t_j)^{\alpha} \to 0,
$$

when $|\Delta_K[0,\infty]| \to 0$, where we have used the fact that $H_{\mathbb{G}}(t)$ is a $\mathbb{P}_{\mathbb{G}}$-martingale and Assumptions 1, 2.

We now proceed to the case when $Q = Q_{\mathbb{G}}$. Indeed, as the limit and expectation can interchange, we can show that

$$
\begin{aligned}
&\left| \mathbb{E}\left[ \Xi_{\mathrm{trt}, \Delta_K[0,\infty]}(H, Q_{\mathbb{G}}) \right] \right| \\
&= \left| \mathbb{E}\left( \sum_{j=0}^{K} \left\{ Q_{\mathbb{G}}(t_j) H(t_j) - Q_{\mathbb{G}}(t_{j-1}) \int H(t_j) \mathbb{G}(\mathrm{d}\bar{a}(t_j) | \bar{A}(t_{j-1})) \right\} \right) \right. \\
&\qquad \left. - \mathbb{E}\left\{ Q_{\mathbb{G}}(0) H(0) - \int H(0) \mathbb{G}(\mathrm{d}\bar{a}(0)) \right\} \right| \\
&= \left| \mathbb{E}\left( \sum_{j=0}^{K} \left\{ Q_{\mathbb{G}}(t_j) H(t_j) - Q_{\mathbb{G}}(t_{j-1}) \int H(t_j) \mathbb{G}(\mathrm{d}\bar{a}(t_j) | \bar{A}(t_{j-1})) \right\} \right) + 0 \right| \\
&\leq \sum_{j=0}^{K} \left| \mathbb{E}\left\{ Q_{\mathbb{G}}(t_j) H(t_j) - Q_{\mathbb{G}}(t_{j-1}) \int H(t_j) \mathbb{G}(\mathrm{d}\bar{a}(t_j) | \bar{A}(t_{j-1})) \right\} \right| \\
&= \sum_{j=0}^{K} \left| \mathbb{E}\left\{ Q_{\mathbb{G}}(t_{j-1}) \mathbb{E}_{\mathbb{G}}[H(t_j) | \mathcal{G}_{t_{j-1}}] \right. \right. \\
&\qquad \left. \left. - Q_{\mathbb{G}}(t_{j-1}) \int H(t_j) \mathbb{G}\{\mathrm{d}\bar{a}(t_j) | \bar{A}(t_{j-1})\} \, \mathbb{P}(\mathrm{d}\bar{l}(t_j) | \mathcal{G}_{t_{j-1}}) \right\} \right| \\
&\leq \kappa \sum_{j=0}^{K} \| H(t_{j-1}) Q_{\mathbb{G}}(t_{j-1}) \|_1 (t_j - t_{j-1})^{\alpha} \\
&\leq \kappa \sup_{t} \| H(t) Q_{\mathbb{G}}(t) \|_1 \sum_{j=0}^{K} (t_j - t_{j-1})^{\alpha} \to 0,
\end{aligned}
$$

when $|\Delta_K[0,\infty]| \to 0$, where we have used the fact that $Q_{\mathbb{G}}(t)$ is a $\mathbb{P}$-martingale and Assumptions 1, 2.

## C.5 PROOF OF THEOREM 4

We first simplify our setting by ignoring censoring and absorbing event time $T$ into $\bar{L}$ as well. This is because the conditional independent censoring assumption (Assumption 2) is known to be nonparametric. Our observed data become $(\bar{A}, \bar{L})$ and full data become $(\bar{A}, \bar{L}_{\mathcal{A}})$.

We first proves that Assumption 1 does not have restrictions on the observed data. We proceed with a constructive proof. For any partition $\Delta_K[0,\infty]$, we define a measure on the full data path space. In fact, one has the knowledge on the decomposition

$$
\mathbb{P}(\mathrm{d}\bar{a}\mathrm{d}\bar{l}) = \prod_{j=0}^{K-1} \left\{ \mathbb{P}(\mathrm{d}\bar{a}(t_{j+1}) | \mathscr{F}_{t_j}) \, \mathbb{P}(\mathrm{d}\bar{l}(t_{j+1}) | \bar{a}(t_{j+1}), \mathscr{F}_{t_j}) \right\} \tag{57}
$$

$$
= \prod_{j=0}^{K-1} \left\{ \mathbb{P}(\mathrm{d}\bar{a}(t_{j+1}) | \bar{a}(t_j), \bar{l}_{\bar{a}}(t_j)) \, \mathbb{P}(\mathrm{d}\bar{l}_{\bar{a}}(t_{j+1}) | \bar{a}(t_{j+1}), \bar{l}_{\bar{a}}(t_j)) \right\}. \tag{58}
$$

Intuitively $\mathbb{P}(\mathrm{d}\bar{l}_{\bar{a}}(t_{j+1}) | \bar{a}(t_{j+1}), \bar{l}_{\bar{a}}(t_j))$ are close to $\mathbb{P}(\mathrm{d}\bar{l}_{\bar{a}}(t_{j+1}) | \bar{l}_{\bar{a}}(t_j))$, whereas the other term $\mathbb{P}(\mathrm{d}\bar{a}(t_{j+1}) | \bar{a}(t_j), \bar{l}(t_j))$ is close to $\mathbb{P}(\mathrm{d}\bar{a}(t_{j+1}) | \bar{a}(t_j), \bar{l}_{\mathcal{A}})$. Therefore, one may define a measure on the full data path space by

$$
\mathbb{P}^F_{\Delta_K[0,\infty]}(\mathrm{d}\bar{l}_{\bar{a}}) := \prod_{j=0}^{K-1} \mathbb{P}(\mathrm{d}\bar{l}_{\bar{a}}(t_{j+1}) | \bar{a}(t_{j+1}), \bar{l}_{\bar{a}}(t_j)) = \prod_{j=0}^{K-1} \mathbb{P}(\mathrm{d}\bar{l}_{\bar{a}}(t_{j+1}) | \bar{a}(t_{j+1}), \mathscr{F}_{t_j}). \tag{59}
$$

Then without loss of generality one may construct $\mathbb{P}^F_{\Delta_K[0,\infty]}(\bar{l}_\mathcal{A})$ by assuming joint independence among $\bar{l}_\mathcal{A}$. One can also define

$$\mathbb{P}^F_{\Delta_K[0,\infty]}(\mathrm{d}\bar{a}|\bar{l}_\mathcal{A}) := \prod_{j=0}^{K-1} \mathbb{P}(\mathrm{d}\bar{a}(t_{j+1})|\mathscr{F}_{t_j}). \tag{60}$$

Then for any sequences of partitions with the mesh going to zero, one may construct a sequence of measures and show that this sequence of measures is Cauchy by a triangular inequality and Assumption 1, following a similar logic as previous proofs. Therefore the sequence converge to a measure $\mathbb{P}^F$, which is independent of the choice of partitions.

Next we need to show that $\mathbb{P}^F$ induces $\mathbb{P}$ on the observed data and $\mathbb{P}^F$ satisfies Assumption 1. The first is trivial because any $\mathbb{P}^F_{\Delta_K[0,\infty]}$ induces $\mathbb{P}$ on the observed data, then so is their limit. To prove the second, for any time $t$ and $\varepsilon > 0$, one might smartly choose a partition $\Delta_K[0,\infty]$ with $\mathbb{P}^F_{\Delta_K[0,\infty]}$ close enough to $\mathbb{P}^F$ and $t, t+\eta \in \Delta_K[0,\infty]$. This can be done because the convergence point is independent of the choice of partitions. We have

$$\mathbb{E}^F(\|\mathbb{P}^F(\mathrm{d}\bar{l}_\mathcal{A}|\mathscr{F}_t) - \mathbb{P}^F[\mathrm{d}\bar{l}_\mathcal{A}|\bar{A}(t+\eta), \mathscr{F}_t]\|_{\mathrm{TV}})$$

$$\leq \mathbb{E}^F(\|\mathbb{P}^F(\mathrm{d}\bar{l}_\mathcal{A}|\mathscr{F}_t) - \mathbb{P}^F_{\Delta_K[0,\infty]}(\mathrm{d}\bar{l}_\mathcal{A}|\mathscr{F}_t)\|_{\mathrm{TV}})$$

$$+ \mathbb{E}^F\{\|\mathbb{P}^F_{\Delta_K[0,\infty]}(\mathrm{d}\bar{l}_\mathcal{A}|\mathscr{F}_t) - \mathbb{P}^F_{\Delta_K[0,\infty]}[\mathrm{d}\bar{l}_\mathcal{A}|\bar{A}(t+\eta), \mathscr{F}_t]\|_{\mathrm{TV}}\}$$

$$+ \mathbb{E}^F\{\|\mathbb{P}^F_{\Delta_K[0,\infty]}[\mathrm{d}\bar{l}_\mathcal{A}|\bar{A}(t+\eta), \mathscr{F}_t] - \mathbb{P}^F[\mathrm{d}\bar{l}_\mathcal{A}|\bar{A}(t+\eta), \mathscr{F}_t]\|_{\mathrm{TV}}\}$$

$$\leq 2\|\mathbb{P}^F - \mathbb{P}^F_{\Delta_K[0,\infty]}\|_{\mathrm{TV}}$$

$$+ \mathbb{E}^F\{\|\mathbb{P}^F_{\Delta_K[0,\infty]}(\mathrm{d}\bar{l}_\mathcal{A}|\mathscr{F}_t) - \mathbb{P}^F_{\Delta_K[0,\infty]}[\mathrm{d}\bar{l}_\mathcal{A}|\bar{A}(t+\eta), \mathscr{F}_t]\|_{\mathrm{TV}}\}$$

$$\leq 4\|\mathbb{P}^F - \mathbb{P}^F_{\Delta_K[0,\infty]}\|_{\mathrm{TV}}$$

$$+ \mathbb{E}^F_{\Delta_K[0,\infty]}\{\|\mathbb{P}^F_{\Delta_K[0,\infty]}(\mathrm{d}\bar{l}_\mathcal{A}|\mathscr{F}_t) - \mathbb{P}^F_{\Delta_K[0,\infty]}[\mathrm{d}\bar{l}_\mathcal{A}|\bar{A}(t+\eta), \mathscr{F}_t]\|_{\mathrm{TV}}\}.$$

The first term can be chosen to be sufficiently small. We rewrite the second term

$$\mathbb{E}^F_{\Delta_K[0,\infty]}\{\|\mathbb{P}^F_{\Delta_K[0,\infty]}(\mathrm{d}\bar{l}_\mathcal{A}|\mathscr{F}_t) - \mathbb{P}^F_{\Delta_K[0,\infty]}[\mathrm{d}\bar{l}_\mathcal{A}|\bar{A}(t+\eta), \mathscr{F}_t]\|_{\mathrm{TV}}\}$$

$$= \sup_{f:\|f\|_1=1} \int f(\bar{l}_\mathcal{A})\{\mathbb{P}^F_{\Delta_K[0,\infty]}(\mathrm{d}\bar{l}_\mathcal{A}|\mathscr{F}_t)$$

$$- \mathbb{P}^F_{\Delta_K[0,\infty]}[\mathrm{d}\bar{l}_\mathcal{A}|\bar{a}(t+\eta), \mathscr{F}_t]\} \, \mathbb{P}(\mathrm{d}\bar{a}(t+\eta)\mathrm{d}\mathscr{F}_t)$$

$$\leq \sup_{f:\|f\|_1=1} \sup_{g:\|g\|_1=1} \int f(\bar{l}_\mathcal{A})g(\bar{a}(t+\eta))[\mathbb{P}^F_{\Delta_K[0,\infty]}[\mathrm{d}\bar{l}_\mathcal{A}\mathrm{d}\bar{a}(t+\eta)|\mathscr{F}_t]$$

$$- \mathbb{P}^F_{\Delta_K[0,\infty]}(\mathrm{d}\bar{l}_\mathcal{A}|\mathscr{F}_t) \, \mathbb{P}^F_{\Delta_K[0,\infty]}\{\mathrm{d}\bar{a}(t+\eta)|\mathscr{F}_t\}] \, \mathbb{P}(\mathrm{d}\mathscr{F}_t)$$

$$= \sup_{f:\|f\|_1=1} \sup_{g:\|g\|_1=1} \int f(\bar{l}_\mathcal{A})g(\bar{a}(t+\eta)) \, \mathbb{P}^F_{\Delta_K[0,\infty]}(\mathrm{d}\bar{l}_\mathcal{A}|\mathscr{F}_t)$$

$$\{\mathbb{P}^F_{\Delta_K[0,\infty]}[\mathrm{d}\bar{a}(t+\eta)|\bar{a}(t), \bar{l}_\mathcal{A}] - \mathbb{P}^F_{\Delta_K[0,\infty]}[\mathrm{d}\bar{a}(t+\eta)|\mathscr{F}_t]\} \, \mathbb{P}(\mathrm{d}\mathscr{F}_t)$$

$$= \sup_{f:\|f\|_1=1} \sup_{g:\|g\|_1=1} \int f(\bar{l}_\mathcal{A})g(\bar{a}(t+\eta)) \, \mathbb{P}^F_{\Delta_K[0,\infty]}(\mathrm{d}\bar{l}_\mathcal{A}|\mathscr{F}_t)$$

$$\left[ \prod_{j=0}^{K-1} \mathbb{P}(\mathrm{d}\bar{a}(t+\eta)|\mathscr{F}_t) - \mathbb{P}(\mathbb{1}) \right] \mathbb{P}(\mathrm{d}\mathscr{F}_t)$$

$$\leq \mathbb{E} \left\| \prod_{j=0}^{K-1} \mathbb{P}(\mathrm{d}\bar{a}(t+\eta)|\mathscr{F}_t) - \mathbb{P}(\mathrm{d}\bar{a}(t+\eta)|\mathscr{F}_t) \right\|_{\mathrm{TV}}$$

$$\leq \varepsilon(t, \eta).$$

Assumption 3 is irrelevant here because it is imposed on the stochastic process but not the distributions.

For Assumption 4, a necessary condition for it to hold is that $\mathbb{P}$ is well supported on the path space conditioning on any filtration. For this to happen, note that for any $\mathbb{P}$, one can find a well-supported $\mathbb{P}'$ satisfying Assumptions 1, 2, 3, so that $(1 - \varepsilon)\mathbb{P} + \varepsilon\mathbb{P}'$ is well-supported and hence satisfies Assumption 4. Since addition will not break Assumptions 1, 2, 3, we have found $(1 - \varepsilon)\mathbb{P} + \varepsilon\mathbb{P}'$ satisfying Assumptions 1, 2, 3, and 4, and approximates $\mathbb{P}$.

# D  ADDITIONAL EXPERIMENT RESULT

In this section, we employ Monte Carlo simulations to empirically assess how the identification works. To that end, we need to go through 4 steps:

1. Come up with a reasonable data generating process;

2. Compute the parameter of interest (1) (or equivalently, the left-hand side of g-computation formula in Theorem 1) according to this data generating process;

3. Simulate according to this data generating process;

4. Approximate the right-hand side of g-computation formula in Theorem 1 using the simulated data.

**Step 1**: For $t \in [0, 1]$, consider a potential outcome process $Y_{\bar{a}}(t)$ and potential covariate $L_{\bar{a}}(t)$ following a Gaussian process with mean process as

$$\begin{pmatrix} \mathbb{E}\{Y_{\bar{a}}(t)\} \\ \mathbb{E}\{L_{\bar{a}}(t)\} \end{pmatrix} = \begin{pmatrix} -a(t) \\ 0.5a(t) \end{pmatrix}, \tag{61}$$

and covariance process

$$\begin{pmatrix} \mathrm{Cov}\{Y_{\bar{a}}(t), Y_{\bar{a}}(s)\} & \mathrm{Cov}\{Y_{\bar{a}}(t), L_{\bar{a}}(s)\} \\ \mathrm{Cov}\{L_{\bar{a}}(t), Y_{\bar{a}}(s)\} & \mathrm{Cov}\{L_{\bar{a}}(t), L_{\bar{a}}(s)\} \end{pmatrix} = \begin{pmatrix} e^{-3|t-s|}/2 & e^{-1-6|t-s|}/2 \\ e^{-1-6|t-s|}/2 & e^{-3|t-s|}/2 \end{pmatrix}. \tag{62}$$

This ensures the joint dependence among $Y_{\bar{a}}(t)$ and $L_{\bar{a}}(t)$ and non-zero treatment effect of $\bar{a}$. Generate the event time following a Cox model

$$\mathbb{P}(T_{\bar{a}} > t | \bar{Y}_{\bar{a}}, \bar{L}_{\bar{a}}) = \exp\left\{-t \exp\left[0.2a(t) - 0.8Y_{\bar{a}}(t) - 2L_{\bar{a}}(t)\right]\right\}. \tag{63}$$

and an independent censoring process

$$\mathbb{P}(C > t) = \begin{cases} \exp(-t), & \text{when } t \leq 1 \\ 0, & \text{when } t > 1, \end{cases} \tag{64}$$

where time 1 represents an administrative censoring. Define $\nu(T_{\bar{a}}, \bar{Y}_{\bar{a}})$ as the integral of $\bar{Y}_{\bar{a}}$ over time $t \in [0, 1]$, that is,

$$\nu(T_{\bar{a}}, \bar{Y}_{\bar{a}}) = \int_0^{T_{\bar{a}} \wedge 1} Y_{\bar{a}}(t)dt. \tag{65}$$

where $T_{\bar{a}}$ is capped at 1 because beyond that all subjects are censored. Suppose the targeted treatment regime $\mathbb{G}$ is a Gaussian measure with mean zero process and jointly independent normal variables at any time points. That is, the intervened $A$ follows a Gaussian process with a mean process

$$\mathbb{E}(A(t)) = 0, \tag{66}$$

and covariance process

$$\mathrm{Cov}\{A(t), A(s)\} = e^{-3|t-s|}/2, \ \forall t, s \in [0, 1]. \tag{67}$$

**Step** : Then we can show that (1) (or equivalently, the left-hand side of g-computation formula in Theorem 1) equals zero, that is,

$$\int \mathbb{E}(\nu(T_{\bar{a}}, \bar{Y}_{\bar{a}}))\mathbb{G}(\mathrm{d}\bar{a}) = \int \mathbb{E}\left[\int_0^{T_{\bar{a}} \wedge 1} Y_{\bar{a}}(t)dt\right]\mathbb{G}(\bar{a}) \tag{68}$$

$$= \int \mathbb{E}\left[\int_0^{\mathbb{E}(T_{\bar{0}}) \wedge 1} Y_{\bar{a}}(t)dt\right]\mathbb{G}(\bar{a}) \tag{69}$$

$$= \int \int_0^{\mathbb{E}(T_{\bar{0}}) \wedge 1} \mathbb{E}(\bar{Y}_{\bar{a}}(t))dt\mathbb{G}(\bar{a}) = \int \int_0^{\mathbb{E}(T_{\bar{0}})} a(t)dt\mathbb{G}(\bar{a}) \tag{70}$$

$$= \int_0^{\mathbb{E}(T_{\bar{0}}) \wedge 1} \int a(t)\mathbb{G}(\bar{a}(t))dt = \int_0^1 0dt = 0. \tag{71}$$

We used the fact that, through our data generating process, the treatment does not have effect on survival time.

**Step 3**: In practice, we observe a stochastic process at finite points. We consider evenly splitting $t \in [0, 1]$ into a grid of size $K + 1$: $\Delta_K[0, 1] = \{t_0 = 0, t_1 = 1/K, \cdots, t_{K-1} = (K-1)/K, t_K = 1\}$, and for $1 \leq i \leq n$, according to $\mathbb{G}$ specified in Step 1, we simulate i.i.d. samples $A_i(t)$ according to $\mathbb{G}$ specified in Step 1 at $\Delta_K[0, 1]$ as

$$\begin{pmatrix} A_i(t_0) \\ A_i(t_1) \\ \cdots \\ A_i(t_{K-1}) \\ A_i(t_K) \end{pmatrix} \sim \mathcal{N}\left[\begin{pmatrix} t_0 - 0.5 \\ t_1 - 0.5 \\ \cdots \\ t_{K-1} - 0.5 \\ t_K - 0.5 \end{pmatrix}, \begin{pmatrix} 1 & e^{-3|t_1-t_0|} & \cdots & e^{-3|t_{K-1}-t_0|} & e^{-3|t_K-t_0|} \\ e^{-|t_1-t_0|} & 1 & \cdots & e^{-3|t_{K-1}-t_1|} & e^{-3|t_K-t_1|} \\ \cdots & \cdots & \cdots & \cdots & \cdots \\ e^{-3|t_{K-1}-t_0|} & e^{-3|t_{K-1}-t_1|} & \cdots & 1 & e^{-3|t_K-t_{K-1}|} \\ e^{-3|t_K-t_0|} & e^{-|t_K-t_1|} & \cdots & e^{-3|t_K-t_{K-1}|} & 1 \end{pmatrix}\right].$$

By according to the distribution of $Y_{\bar{a}}(t)$ specified in Step 1 and consistency, we generate $Y_i(t)$ and $L_i(t)$ following a similar manner. Generate the event time following a Cox model

$$\mathbb{P}(T_i > t | \bar{A}_i, \bar{Y}_i, \bar{L}_i) = \exp\left\{-t \exp\left[0.2A(t) - 0.8Y_i(t) - 2L_i(t)\right]\right\}. \tag{72}$$

and an independent censoring process

$$\mathbb{P}(C_i > t) = \begin{cases} \exp(-t), & \text{when } t \leq 1 \\ 0, & \text{when } t > 1, \end{cases} \tag{73}$$

**Step 4**: The integral of $Y_i(t_k)$ over $[0, 1]$ is $\sum_{0 \leq k \leq T_i} Y_i(t_k)/(K+1)$. The approximate of the right-hand side of g-computation formula is $\sum_{i=1}^n \sum_{0 \leq k \leq T_i} Y_i(t_k)/(K+1)/n$.

We vary the grid sizes ($K = 10, 50, 250$) to examine how a denser grid improves the approximation. This approach simulates the scenario where the mesh $|\Delta_K[0, 1]|$ is shrunk to zero. Additionally, we vary the sample sizes ($n = 100, 500, 2500$) to explore how larger samples enhance the approximation, leveraging the law of large numbers to better approximate the right-hand side of the g-computation formula. We repeat the process $R = 10,000$ times. The resulting 10,000 approximations of $\sum_{i=1}^n \sum_{k=0}^K Y_i(t_k)/(K+1)/n$ are presented in boxplots in Figure D, where we append biases.

The simulation results demonstrate that the g-computation formula can adequately approximate (1) even with moderate sample and grid sizes. The biases are larger than that in Section 4 possibly because we are focusing on a more difficult task. This time, however, increasing the sample size while keeping the grid size fixed does not enhance the accuracy. Increasing the grid size while keeping the sample size fixed does improve accuracy or reduce variance in this case. Again, simultaneously increasing both the sample and grid sizes significantly improves accuracy and reduces variance in the approximation.

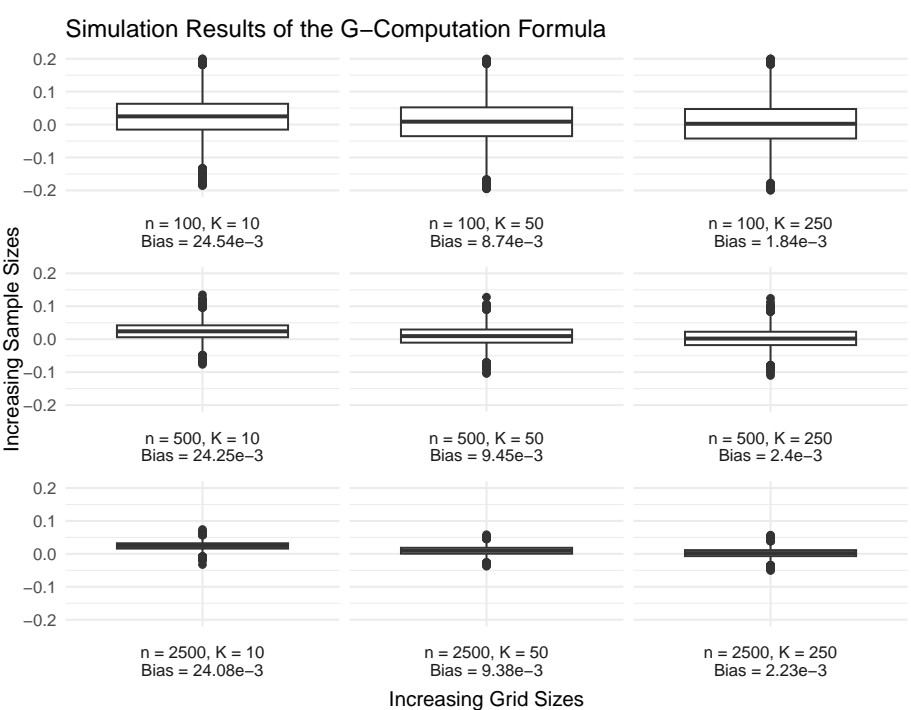

Figure 2: Simulation Results of using g-computation formula by varying grid sizes in $K = 10, 50, 250$ and sample sizes in $n = 100, 500, 2500$, for $R = 10000$ repeats. We plot boxplots and give biases.

