# OpenReview forum: "Causal Identification for Complex Functional Longitudinal Studies"
_ICLR.cc/2025/Conference — ICLR 2025 Poster_

### Official Review · Reviewer_WrGA · 2024-11-02

**Soundness:** 2
**Presentation:** 1
**Contribution:** 2
**Rating:** 3
**Confidence:** 2

**Summary:**

This paper presents a nonparametric causal identification framework for functional longitudinal data, exemplified by the MIMIC-IV dataset, which includes complexities such as events like death. By leveraging stochastic process theory and measure theory, the framework generalizes g-computation, inverse probability weighting, and doubly robust formulas, effectively handling time-varying outcomes with mortality and censoring. Monte Carlo simulations verifies it.

**Strengths:**

This paper addresses the problem of causality identification in complex scenarios inherent in functional longitudinal data, a highly general and advanced form of data. If it can overcome the criticisms and shortcomings pointed out by other reviewers and get published, it could serve as a milestone, taking one step forward in functional data analysis for causal inference.

**Weaknesses:**

This paper is more abstract than necessary, making it harder to read than other papers. Even after spending more time on it than on other review papers, the preliminaries, equations, and theorems presented in the paper are not easily understood at once. As a result, I had to refer to the cited papers, and I found two papers that share many aspects with this one. One is "Causality for Functional Longitudinal Data," which is cited by (Ying, 2024) in this paper, and the other is "Causality for Complex Continuous-Time Functional Longitudinal Studies with Dynamic Treatment Regimes," which is not cited in this paper. The title of this paper is "Causal Identification for Complex Functional Longitudinal Studies," suggesting that the most significant update in this paper is the "complex" aspect in the title. The paper "Causality for Complex Continuous-Time Functional Longitudinal Studies with Dynamic Treatment Regimes" seems to have updated the dynamic treatment regimes element one step further.

Specifically, in the preparation section, the time-to-event endpoint $T$, $C$, $X$, and $\Delta$ are newly defined, assuming a more complex situation where the study may be forcibly terminated due to an event like death before the study is completed. This updates Theorems 1, 2, and 3 from the previous paper. I would like to ask about the academic and practical significance of solving the more complex problem introduced by those new variables. The most important contribution of this paper seems to be the introduction of the non-parametric property through Theorem 4. Why is the non-parametric property important in a functional data framework? The paper states that it makes the model more flexible and adaptable to various data, but doesn't the continuous functional data, which is more extensive, lead to increased computational and implementation complexity, a common drawback of non-parametric models? Doesn't this also create issues with the interpretability required for healthcare data analysis?

One of the most important equations in this paper seems to be Equation (1). The rest of the paper is dedicated to finding another representation of Equation (1). However, it is not easy for readers to immediately understand what Equation (1) means and why we should be interested. Additionally, it is not straightforward to grasp what $\mathbb{G}$ represents. By referring to the paper by Ying (2024), I could somewhat understand $\mathbb{G}$ through the following example:
*When the causal outcome under a specific regime $\bar{a}$ is of interest, for instance, all patients were under treatment, the point mass (delta) measure $\mathbb{G} = \mathbb{1}(\bar{A} = \bar{a})$ can be considered.*
Including this example in this paper would help in understanding. Furthermore, providing a concrete example of what $\nu$ represents would help comprehend Equation (1). Can Equation (1) be understood as a general expression representing the average treatment effect, the averaged treatment outcome, or a transformed form of these?

If the non-parametric property is a major contribution of the paper, it should be demonstrated through more concrete experimental examples, such as using the MIMIC-IV data mentioned in the introduction. The experimental section currently numerically verifies Theorem 1, which has already been proven in the Appendix, but a demonstration of Theorem 4 seems more necessary. However, under Theorem 4, it is only mentioned that "we have not achieved the full nonparametric paradigm."

Additionally, there is a need to clearly and definitively define loosely defined “functional” data. The abstract of this paper describes it as "characterized by continuous-time measurements," while another cited paper describes it as "characterized by continuous-time processes and high-dimensional measurements." I believe "continuous" alone is not sufficient to be called functional. What is the rationale for developing a framework that assumes functional continuity in the model even though real-world healthcare data does not have mathematically rigorous time continuity and does not observe over an infinite time? (The previous paper assumed up to time $\tau$, but this paper assumes up to $\infty$.) What is the justification for this assumption?

**Questions:**

Please provide additional explanations for the questions raised in the Weakness section.

**Details Of Ethics Concerns:**

Figure 1 is directly taken from Figure 2 of the paper "Causality for Complex Continuous-Time Functional Longitudinal Studies with Dynamic Treatment Regimes" submitted to the Annals of Statistics (https://arxiv.org/pdf/2406.06868). A citation should be included.

---

> ### Author Response · Authors · 2024-11-15
>
> Thank you for your detailed and constructive feedback, and for recognizing the strengths of this work. We appreciate the time you dedicated to our manuscript and the opportunity to address your concerns. Below, we respond to the key points you raised.
>
> ### Weaknesses
>
> 1. **Relation to "Causality for Complex Continuous-Time Functional Longitudinal Studies with Dynamic Treatment Regimes":**
>    We acknowledge the connection with this paper, which indeed represents a subsequent work to the present one. While the later work generalizes identification results to include semiparametric frameworks, it lacks numerical examples and practical verifications. Our current submission provides a focused exploration of nonparametric identification, coupled with simulation-based numerical validation. Incremental contributions like this allow for a deeper understanding of specific aspects, offering clarity and actionable insights while building toward more comprehensive frameworks.
>
> 2. **Academic and Practical Significance of Solving More Complex Problems:**
>    The introduction of variables such as \(T\), \(X\), and \(\Delta\) addresses complexities inherent in real-world medical applications. Medical studies often involve truncated follow-up due to death or censoring, which these variables explicitly model. Furthermore, in healthcare, interest frequently lies in the entire process of progression (e.g., \(Y(t)\)), rather than just a terminal outcome. By distinguishing between these components, our framework supports broader and more clinically relevant causal queries.
>
> 3. **Why is the Non-Parametric Property Important in a Functional Data Framework?**
>    The non-parametric property aligns with the recent assumption-lean efforts in the causal inference community (see references below). Traditional approaches often rely on parametric or semi-parametric modeling assumptions, such as smoothness or sparsity, to facilitate analysis and reduce dimensionality. However, these assumptions are typically made for mathematical convenience rather than being grounded in prior knowledge. As a result, inferences drawn from such models may reflect the assumptions as much as, or more than, the data itself.
>
>    For functional data, where the complexity of continuous, infinite-dimensional outcomes makes it even harder to justify any specific model, relying on parametric assumptions becomes especially unrealistic. Our framework deliberately separates modeling assumptions from identification, focusing purely on structural assumptions necessary for causal inference. This ensures that the framework extracts information only from the data, avoiding the risk of introducing unwarranted or misleading conclusions based on arbitrary assumptions.
>
>    By adopting a non-parametric approach, we provide a more flexible and adaptable methodology that reflects the complexities of real-world data, particularly in healthcare settings where data rarely adhere to idealized models. This choice strengthens the framework’s robustness and relevance, particularly for functional data analysis.
>
>    References:
>    - Vansteelandt, Stijn, and Oliver Dukes. "Assumption-lean inference for generalised linear model parameters." *Journal of the Royal Statistical Society Series B: Statistical Methodology* 84.3 (2022): 657-685.
>    - Vansteelandt, S., Dukes, O., Van Lancker, K., & Martinussen, T. (2024). Assumption-lean Cox regression. *Journal of the American Statistical Association,* 119(545), 475-484.
>
> 4. **More explanations on \(v\) and \(G\):**
>    Yes, we are adding more explanations and examples around them. Thank you for pointing this out!
>
> 5. **"Can Equation (1) be understood as a general expression representing the average treatment effect, the averaged treatment outcome, or a transformed form of these?"**
>    Yes, our characterization can accommodate all the cases you’ve mentioned here because we’ve allowed \(G\) to be a signed measure. This means it does not have to be positive, allowing, for instance, \(1(\bar{A} = \bar{1}) - 1(\bar{A} = \bar{0})\) (the difference of two delta measures), which represents the average treatment effect of always-treated vs. never-treated.

---

> ### Author Response · Authors · 2024-11-15
>
> 5. **Demonstration of Theorem 4**
>    Thank you for highlighting the importance of Theorem 4. We agree that it is a key contribution of the paper, and we understand the desire for its further validation. However, it is important to note that Theorem 4 establishes a theoretical foundation rather than a practical computation. Specifically, the theorem demonstrates that our framework is dense in the probability measure space, which implies that in real-world scenarios, one cannot practically encounter a measure that violates it. As such, there is no established practice or meaningful way to numerically verify this property.
>
>    Instead, Theorem 4 serves as the cornerstone for future work, ensuring that any estimation procedure—whether parametric or non-parametric for nuisance functions—can operate freely within our framework. This generality is crucial for enabling subsequent developments in functional data analysis for causal inference, including estimation, inference, and applications to real-world datasets like MIMIC-IV.
>
>    While we acknowledge the importance of practical validation, the contribution of Theorem 4 is primarily theoretical, laying the groundwork for future empirical applications and methodological advancements.
>
> 6. **Definition of "Functional" Data**
>    Thank you for your observation regarding the definition of "functional" data. We will incorporate the feedback from other reviewers and refine our definition, drawing from established references such as Wang, Chiou, and Müller (2016). Specifically, we will clarify that functional data refers to data arising from continuous-time processes, often characterized by high-dimensional, smooth trajectories or curves, and distinguish our focus within this framework. This will help to more rigorously position our approach while emphasizing its relevance to causal inference in complex scenarios.
>
> 7. **Assumption of Functional Continuity**
>    The assumption of functional continuity reflects the population-level nature of our framework. The causal effect of a drug, for instance, operates in a continuous-time manner, even if real-world data are observed discretely. How sample-level data are recorded—whether densely, sparsely, or irregularly—is a topic for future work focusing on estimation. These considerations will include the minimum observation density required for consistent estimators and the potential shift to partial identification when non-dense observations are present.
>
>    In this paper, our focus is on rigorously defining the estimand in a way that reflects the underlying continuous process at the population level. This definition is essential for future developments in estimation and inference, ensuring the framework remains robust and grounded in theoretical principles.
>
>    For ethical concerns, Figure 1 will be removed according to another reviewer's suggestion to make room for more explanation.

---

> > ### Comment · Reviewer_WrGA · 2024-11-26
> >
> > Thank you for your response. The additional explanations and materials helped me understand some questions better. However, the paper's readability is still low and remains ambiguous. While I cannot point out everything, for example, the equations from (2) to (13) are very complex and messy. Can they be simplified?
> >
> > You explained the connection with other papers above, but since the discussion period has been extended, I would like to ask again for clarity:
> > Could you briefly and clearly summarize what new additions have been made compared to the following two papers:
> > - "Causality for Complex Continuous-Time Functional Longitudinal Studies with Dynamic Treatment Regimes"
> > - "Causality for Functional Longitudinal Data"
> >
> > Additionally, could you please explain the practical significance and substantial contributions of these new additions? The methodologies used seem to be substantially shared with those papers, and it appears that the use of newly added variables (e.g., $T_{\bar{a}}$) has slightly expanded the scope. Therefore, judging the meaning of these additions and contributions is still somewhat unclear. A brief and precise summary of the substantial contributions would help make a final judgment.
> >
> > One fundamental question I would like to ask again is, as you roughly explained above, real-world clinical data is far from mathematical rigor, such as continuity and infinity. Why is it necessary to develop mathematically rigorous models or frameworks? My point is, how can this theoretically developed framework be persuasive in its importance and necessity when it cannot be verified with real-world data? I am unsure that the MIMIC-IV data is appropriate as an example of continuity. It sounds like claiming that quantum mechanics and general relativity will be needed in the future to analyze data in the Newtonian era. Judging a paper that lacks, as you agreed and explained above, established practices or meaningful ways to verify its core contributions is unclear. At this point, it feels more appropriate for a specialized statistical mathematics journal rather than ICLR, which might seek papers with immediate impact.

---

> > > ### Author Response · Authors · 2024-11-26
> > >
> > > Thank you for your thoughtful questions. I want to give a quick answer for your fundamental question (I am working on the first few questions).
> > >
> > > While clinical data is often recorded discretely, the underlying processes, such as disease progression or physiological responses, are continuous. This parallel is seen in the development of continuous-time models like the Cox and Aalen models, which provide deeper insights into survival dynamics despite being applied to discretely measured data. For example, many patient datasets are recorded on a monthly basis yet have not stopped the development of these continuous-time models.
> > >
> > > The use of mathematical rigor, including concepts like limits and integration, relies on idealized notions of infinity while real-world data is finite and discrete. However, these abstractions enable better approximations, deeper understanding, and more general frameworks. For example, calculus itself, which underpins nearly all scientific advancements, emerged from approximating real-world phenomena through continuous and infinite concepts.
> > >
> > > Furthermore, there is a growing interest within the ICLR community in machine learning for functional data. By bridging the gap between causal inference and functional data analysis, our work provides the foundation for future estimation frameworks that functional data experts can build upon. This conference offers the ideal platform to engage these researchers, fostering collaboration and inspiring practical advances that illustrate the impact of this work.

---

> > > ### Author Response · Authors · 2024-11-27
> > >
> > > We tried our best to rewrite (2) - (13) is a looser but simpler manner. We've move the old and rigorous one into the appendix.

---

> > > ### Author Response · Authors · 2024-11-28
> > >
> > > Thank you once again for your detailed and thoughtful review of our paper. I’m especially grateful for your recognition of the paper’s potential to be a milestone in advancing functional data analysis for causal inference in the Strengths section, which is both encouraging and motivating.
> > >
> > > Since your initial review, I have worked diligently to address the concerns raised by all reviewers, including the specific criticisms and shortcomings you referenced. I am pleased to share that the other reviewers have acknowledged these efforts, reflected in their updated scores.
> > >
> > > Given the significant progress made in addressing these concerns, I kindly ask if you might consider revisiting your score in light of these updates. Your appreciation of the importance of this work and any further guidance you could offer would mean a great deal to me.
> > >
> > > Thank you for your time and continued engagement in the review process.

---

> ### Author Response · Authors · 2024-11-26
>
> I hope you’ve had a chance to review my responses to your comments on our paper. Please let me know if there are any additional concerns.

---

> ### Author Response · Authors · 2024-11-27
>
> We totally agree that the connection with these two papers can be made even clearer.
>
> * **Additions to "Causality for Functional Longitudinal Data" (referred to as "Ying (2024a)")**:
>   * Ying (2024a) served as an initial stepping stone pushing the boundaries of causal inference into "functional longitudinal data," focusing on an oversimplified setting: a single, end-of-study outcome measured at one point in time.
>   * In contrast, our paper, **as an extension pushing the boundary towards more realistic data**, introduces not only the newly added variable $T_{\bar{a}}$ but also incorporates a time varying outcome process $\bar Y_{\bar{a}}$. Both are time-varying processes yet they are different because $T_{\bar{a}}$ is a terminating 0-1 process that prevents the observation of other processes, whereas $\bar{Y}_{\bar{a}}$ is truly functional (or curve) data. Furthermore, we allow the data to be subject to right censoring ($C$). While the methodology may seem similar, this progression mirrors how incremental causal inference research has historically evolved—from simpler to more complex scenarios. Confirming the framework's applicability to a more complicated setting is both non-trivial and meaningful.
>   * Our framework showcases nonparametric properties, which Ying (2024a) does not address.
>   * We included an extensive simulation study, which is absent in Ying (2024a).
>
> * **Additions to "Causality for Complex Continuous-Time Functional Longitudinal Studies with Dynamic Treatment Regimes" (referred to as "Ying (2024b)")**:
>   * Ying (2024b) extends the boundaries drawn in our paper to dynamic treatment regimes (DTR), a broader and more complex concept. However, this required substantial preparation, making the paper more difficult to understand. For instance, Ying (2024b) dedicates a full section (Section 3) to defining counterfactual outcomes under DTRs, a challenging extension. In contrast, our paper focuses on static treatment regimes (where the intervened distribution of $\bar{A}$ does not depend on covariates), a cleaner and more natural starting point. This mirrors how initial investigations in other contexts often begin with simpler concepts, such as average treatment effects for point exposures or marginal structural models for regular (discrete-time) longitudinal studies, both of which fall under static treatment regimes.
>   * To prove nonparametric properties, Ying (2024b) relies on semiparametric theory and shows that the tangent space under its assumptions equals $L_0^2(P)$. In contrast, our paper takes a direct approach, proving that the subset of probability measures satisfying our assumptions is dense.
>   * We included an extensive simulation study, which Ying (2024b) does not have.
>
> We hope this summary clarifies how our paper builds upon and differentiates itself from these foundational works.

---

> ### Author Response · Authors · 2024-12-02
>
> The discussion period for reviewers ends on 12/02, please let me any follow up questions and we will be glad to answer.

---

> > ### Comment · Reviewer_WrGA · 2024-12-02
> >
> > Of course, disease progression or physiological processes occur continuously over time, but how is the nature of these continuous processes captured by data measured at weekly or monthly intervals? Especially considering the noise in measurements.
> >
> > The abstract of this paper starts with the following sentence:
> >
> > “Real-time monitoring” in modern medical research introduces functional longitudinal data, characterized by “continuous-time measurements” of outcomes, treatments, and confounders. This complexity leads to uncountably “infinite treatment confounder feedbacks” and “infinite-dimensional data”, which traditional causal inference methodologies cannot handle......
> >
> > Does this paper aim to analyze discretely measured or continuously measured data of continuous processes? Your answer suggests both, but the abstract indicates the latter. I am unsure if modern medical fields produce infinite-dimensional data through continuous monitoring. Is MIMIC-IV data like that? If there is any data I am missing, please let me know.
> >
> > Thank you for comparing the papers. In summary, this paper differentiates itself by addressing more realistic data settings, incorporating nonparametric properties, and focusing on static treatment regimes. Is that correct?
> >
> > I acknowledge your diligent responses and the revisions made to the paper. However, I cannot recommend this paper for acceptance to the area chair. While it may be a good paper for those researching this specific subfield, it is not easily readable for general experts in the field of causal inference in machine learning. I have spent a lot of time reading and trying to understand the paper, and I am providing my questions and reviews, but it is still difficult to evaluate the value of this paper. It is hard to recommend a paper that is not well understood and evaluated. I would like to lower my voice and reduce my confidence score from 3 to 2.

---

> ### Author Response · Authors · 2024-12-02
>
> Thank you for your detailed feedback. I appreciate the time you have taken to read the paper and provide thoughtful comments and questions.
> ---
>
> ### 1. Clarification on Data Type (Continuous vs. Discrete Measurements)
> - **Short Answer:** Our paper focuses on discretely measured data from continuous processes because, in practice, it is impossible to store or analyze infinite data.
>
> - **Long Answer:** In statistical inference, especially in the causal inference framework, there are typically two steps: **identification** and **estimation**:
>   - **Identification** defines the parameter of interest and determines how, given infinite copies of data, one can mathematically identify the parameter. This is the focus of our paper.
>   - **Estimation** involves constructing estimators using finite samples with desirable statistical properties, such as consistency or efficiency. While many studies integrate both steps, our paper specifically focuses on identification for **functional longitudinal data**.
>
> Beyond these two layers, for functional data derived from continuous processes, there is an additional layer: **representation error**. This refers to the gap between having infinite copies of discrete-time observations (what is practical) versus infinite copies of continuous-time observations (theoretical ideal). While representation error is an important aspect, it is tied to the estimation step and is therefore outside the scope of this paper. Note that this is pointed out by reviewer dyEJ's summary.
>
> - Representation error has been well-discussed in works like "Wang, Chiou, and Müller (2016)," which categorizes discrete-time observations into dense, sparse, or irregular regimes. Exploring how representation error diminishes under specific assumptions (e.g., number and frequency of observations) is indeed an interesting direction, but our paper focuses strictly on identification under the framework of **functional longitudinal data**.
>
>
> ### 2. Comparing Papers and Differentiation
> You summarized correctly that our paper differentiates itself by addressing more realistic data settings, incorporating nonparametric properties, and focusing on static treatment regimes. Additionally, I want to highlight a key feature that sets our paper apart: **the inclusion of a numerical study**, which the other papers lack. This provides empirical evidence supporting our theoretical findings and strengthens the practical relevance of our approach.

---

### Official Review · Reviewer_dyEJ · 2024-11-02

**Soundness:** 3
**Presentation:** 1
**Contribution:** 2
**Rating:** 5
**Confidence:** 4

**Summary:**

The paper is challenging to follow, so I may have misunderstood some parts. My understanding is that the "functional longitudinal data" investigated here are conventional functional data, as described by Wang et al. (2016), which can be measured intensively, sparsely, or irregularly. However, this paper focuses solely on the ideal (hypothetical) setting where continuous-time measurements are available for each experimental subject, resulting in infinite-dimensional data. If this interpretation is correct, the goal of this paper is to explore causal identification for infinite-dimensional functional (time-varying) outcomes that are subject to mortality and censoring by generalizing the classical g-computation, inverse probability weighting, and doubly robust formulas.

Reference:
Wang, Chiou and Müller (2016). Functional data analysis. Annual Review of Statistics ands its application.

**Strengths:**

The approach is nonparametric and it accommodates functional treatment processes A(t) and functional confounders L(t), as well as functional response Y(t).

**Weaknesses:**

The paper is hard to follow and the connection of the event-time T to the outcome Y(t) is unclear.

**Questions:**

Could you elaborate on the situation when A(t) is a function?

Why should Y(t) be a subset of L(t), and what does it mean?

---

> ### Author Response · Authors · 2024-11-15
>
> Thank you for your thoughtful and detailed feedback. Your comments provide valuable insights that will help us improve both the presentation and clarity of our manuscript. Below, we address your specific concerns and questions.
>
> Clarification of Summary
>
> We appreciate your summary and would like to clarify that our focus is on the underlying curve data \( X(t) \), as introduced in Wang et al. (2016). At the population level, our framework abstracts away the sparsity or regularity of sample-level observations. In future work, we plan to extend our framework to sample-level data, where factors such as sparsity or irregularity could influence the consistency of estimators. For example, investigating how the number of observed time points \( p_n \) scales with the sample size \( n \) in densely observed data could provide valuable insights.
>
> Weaknesses
>
> 1. Connection Between \( T \) and \( Y(t) \):
>    We will improve the explanation of how \( T \) (the event time) relates to \( Y(t) \) (the outcome). Intuitively, \( T \) serves as a "Cemetery Point" where all other stochastic processes cease to evolve due to the individual’s death. For instance, if \( Y(t) \) represents disease progression and \( T \) represents death, \( Y(t) \) is fixed as \( Y(T) \) for \( t > T \). Both \( Y(t) \) and \( T \) are practically relevant, so we distinguish them in our framework. This distinction aligns with prior work such as Rytgaard et al. (2022).
>
> 2. Improving Readability:
>    We will follow the concrete steps suggested by other reviewers, including simplifying notations, adding examples, and providing more intuitive explanations throughout the manuscript. This will make the connection between key components clearer.
>
> Questions
>
> 1. When \( A(t) \) is a Function:
>    The results in our paper remain unchanged if \( A(t) \) is a function, vector, or scalar. This is because our framework is built using measure theory, which generalizes across these cases. Theorems and results retain their form regardless of the specific nature of \( A(t) \).
>
> 2. Why \( Y(t) \subseteq L(t) \):
> This is purely for notational simplicity. Including \( Y(t) \) separately would make the measures and derivations more cumbersome without adding clarity. The framework does not assume \( Y(t) \) must impact treatment assignment but allows this dependency to exist or not, reflecting scenarios like disease progression influencing treatment decisions.

---

> > ### Comment · Reviewer_dyEJ · 2024-11-25
> >
> > Thanks for confirming that your method currently focuses on curve data. I also appreciate the clarification on other issues and have no further questions.

---

> > > ### Author Response · Authors · 2024-11-25
> > >
> > > Thank you for your comments and we greatly appreciate your feedback, which has been invaluable in improving our manuscript.
> > >
> > > Given the efforts to address the points raised and the potential of our work that you kindly highlighted, we would like to respectfully request that you reconsider your score, if possible. We believe the revisions and planned updates significantly strengthen the paper’s contribution and clarity.

---

> > > > ### Comment · Reviewer_dyEJ · 2024-11-28
> > > >
> > > > I appreciate the discussions so far but still find the paper hard to follow as many notations have not been clearly defined.
> > > > There may be something novel in this work but it is difficult to figure them out in its present form.  Nevertheless, I am willing to give the authors the benefit of doubt and will raise my score from 3 to 5.

---

> > > > > ### Author Response · Authors · 2024-11-28
> > > > >
> > > > > Thank you for your revisit of your scoring. I cannot make any changes to the pdf but the discussion period is extended. During that time, feel free to drop any concerns you have about unclear notation. It will help us in refining the paper either in the camera ready version if accepted or its future form. We will really appreciate it.
> > > > >
> > > > > Thank you for your time and continued engagement in the review process.

---

### Official Review · Reviewer_2NEN · 2024-11-02

**Soundness:** 3
**Presentation:** 3
**Contribution:** 3
**Rating:** 8
**Confidence:** 3

**Summary:**

This paper proposes a causal identification framework that bridges classical causal inference framework, continuous-time longitudinal analysis and functional data analysis. In this framework, the parameter of interest is the marginal mean of counterfactual outcomes under a measure that allows randomly assigned treatments, with absence of censoring. Leveraging the tools in stochastic process, the authors then demonstrate the identification results for three classical estimation strategies in causal inference: g-computation, IPW and doubly robust estimation. The authors further claims that the identification framework also has non-parametric property.

**Strengths:**

This paper establishes a new causal identification framework for continuous-time longitudinal studies with functional data, and provides clear and concise theoretical demonstration. I believe that this framework will be of interest to causal inference and machine learning communities.

**Weaknesses:**

1. The numerical experiment might be an over-simplification of the survival analysis scenario since neither mortality nor censoring are taken into consideration.
2. What is the causal structure that the framework is focusing on? Specifically, why set $Y(t)$ (outcome of interest) to be a subset of $L(t)$ (measured confounders)? I might misunderstood but are we assuming that previous outcome will impact the current treatment assignment (since confounders, from my understanding, will impact treatment assignment)?
3. I guess it would be helpful to attract readers in a wider community if more intuitive explanation could be added after stating definitions/propositions.

**Questions:**

1. Why is the interventional distribution ((7)-(10)) formulated in this way? Specifically, I’m curious about where the term $\{1 -   \mathbb{1}(x \leq t_{j+1}, \delta=0) }$ comes from.
2. Can this framework be extended to dependent censoring?

---

> ### Author Response · Authors · 2024-11-15
>
> Thank you for your constructive feedback and thoughtful questions. We appreciate your recognition of the contributions and potential impact of our work. Below, we address the key points you raised.
>
> Weaknesses
>
> 1. Numerical Experiment Simplification:
>  We acknowledge the limitations of the current simulations and will incorporate additional scenarios accounting for mortality and censoring in the revised manuscript.
>
> 2. Causal Structure and \( Y(t) \subseteq L(t) \):
> This notation was chosen purely for simplicity. Including \( Y(t) \) separately would make the measures and derivations more complex without adding clarity. We are not assuming \( Y(t) \) must affect treatment assignment but instead allow this dependency to exist or not. This flexibility is critical as in many cases (e.g., disease progression), outcomes can influence treatment adjustments, and therefore acting as a confounder as well.
>
> 3. Intuitive Explanations for Propositions:
>  We agree with this suggestion and will include intuitive explanations and examples following definitions and propositions, addressing similar feedback from other reviewers.
>
> Questions
>
> 1. Interventional Distribution (Equations (7)-(10)):
>    We will expand this section to bridge the conceptual gap from Equations (2)-(6) to (7)-(10). Specifically:
>    - Equations (4)-(5) describe the probability of censoring within or beyond \([t_j, t_{j+1}]\).
>    - Intervening to a pseudo-world where censoring always happens after \(t_{j+1}\) leads to terms like \( {1 - 1(x \leq t_{j+1}, \delta = 0)} \).
>    - Similarly, treatment distributions are intervened into \(G\) as shown in Equation (10).
>    For additional context, we will reference Rytgaard et al. (2022) in the reference of our paper, particularly Definitions 1 and 2, to clarify intervention-based causal inference.
>
> 2. Extension to Dependent Censoring:
>    Yes, our framework can be extended to handle dependent censoring. If the dependency is explained by observed factors, this is already addressed under Assumption 2. For unobserved dependency, methods like proxy variables (e.g., Ying, A. (2024). Proximal survival analysis to handle dependent right censoring. Journal of the Royal Statistical Society Series B: Statistical Methodology, qkae037.) can be adapted to extend our framework. Similarly, this approach could be generalized for unmeasured confounders.

---

> > ### Comment · Reviewer_2NEN · 2024-11-26
> >
> > Thank you for the authors' thoughtful response. I'm satisfied with the response. Additionally, I appreciate the authors' inclusion of additional examples and details in the revision. With all these improvements taken into account, I would like to elevate the score.

---

> > > ### Author Response · Authors · 2024-11-26
> > >
> > > Thank you so much for your re-consideration.
> > >
> > > Good news, the more complicated simulation is also finished and added to the appendix.

---

### Official Review · Reviewer_Fipq · 2024-11-04

**Soundness:** 2
**Presentation:** 3
**Contribution:** 2
**Rating:** 5
**Confidence:** 2

**Summary:**

In this paper, authors consider causal inference on time-varying data (functional longitudinal data). They generalize the classical g-computation, inverse probability weighting and doubly robust formulas to the time-varying setting subject to censoring and mortality. The g-computation formula is simulated using Monte-Carlo on a toy dataset, achieving promising results.

**Strengths:**

Treatment effect on functional longitudinal data seems to be an understudied subject. This research nicely fills the gap of existing works.

The resulting G-computation can be quite straightforwardly approximated using observations under simulation settings.

**Weaknesses:**

The way this paper is written obscures its main ideas (at least to a general, non-expert reader). There are many terms and phrases used without clear explanation (e.g., "g-computation", "counterfactual time-to-event endpoint"). This restricts the range of potential readers of this paper.

**The experiments are limited to only simulation data and only validate the G-computation formula**.

The literature review of this paper (section 2) does not seem to provide much information of existing works as without proper explanation, readers may be unclear what "temporal aspect", "point exposure" and "end-of-study outcome" means. I recommend removing Figure 1 and expanding on each of the subsections, providing more details of existing works.

The preparation in Section 3.1 is quite long. Without concrete examples, it is hard for readers to understand what they actually mean. I suggest skip some unnecessary notations, and explain them as the paper progresses.
    - Some symbols are better explained with examples. For instance, authors could give an example of nu and G, around equation (1).

**Questions:**

Line 141, authors mentioned "note this is not a density function". Then please specify what this is.

Line 325, why is it sufficient to evaluate the approximation of the G-computation formula? I don't think on population level, the values of three formulas are numerically equal. Even if they are equal, they may have quite different finite-sample behaviors.

---

> ### Author Response · Authors · 2024-11-15
>
> Thank you for your detailed and thoughtful feedback. Your comments provide valuable guidance for improving our manuscript. Below, we address the key points you raised. The draft is under changing now to incorporate all reviewers' suggestions but here is a preliminary reply for what we will do and also answer some of your questions.
>
> General Revisions
>
> We will implement all suggested improvements in the "Weaknesses" section to enhance clarity and accessibility:
> 1. Terms such as "g-computation" and "counterfactual time-to-event endpoint" will be clearly defined with examples. We will also add reviews of some terms in the classical discrete-time case in the appendices.
> 2. The literature review will be expanded to include detailed textual explanations, replacing Figure 1.
> 3. Section 3.1 will be streamlined with concrete examples (e.g., for 𝜈 and 𝐺) and unnecessary notations removed.
>
> Specific Responses to Weaknesses
> Experiments Limited to Simulation Data:
>
> The primary aim of our paper is to address the identification problem for functional longitudinal data. As you and other reviewers noted, this is a novel area in the field, and our focus in this work is strictly on theoretical identification rather than estimation or inference. Given the 10-page limit and the complexity of the problem, our approach represents an incremental but crucial step in developing a solid foundation for future estimation frameworks. As described in the paper, the g-formula, inverse probability weighting (IPW), and doubly robust (DR) formulas yield identical results at the population level under identification, all equal to (1). Since all three are theoretically equivalent on the population level, verifying the g-formula suffices as a sanity check for our purposes. Additionally, the g-formula is the most computationally feasible for simulation because its implementation does not require knowledge on the measures themselves, unlike IPW or DR (unlike in discrete-time and non-functional cases, one have the knowledge and can compute the density, here we need to compute measures, which is far from trivial). This also addresses Question 2: At the population level, the three formulas are theoretically equivalent under our framework. Their finite-sample differences stem from estimation, which will be a focus of future research.
>
> With that said, we realized that our simulation can be oversimplified, as noted by other reviewers as well. Therefore, we will make our simulations more complex by adding more covariates, separating from the outcomes, adding mortality and censoring as well.
>
> Clarity Regarding the Statement on Density Functions (Line 141):
>
> In infinite-dimensional spaces, the concept of density is not applicable as in finite-dimensional Euclidean spaces. Instead, one must resort to measure-theoretic approaches. For example, the measure 𝑃(𝑑\bar{𝑎}𝑑\bar{𝑙})) referenced in the paper pertains to the probability distribution over the paths of stochastic processes, which is characterized directly via measures rather than densities. We will revise this section to clearly explain why density functions are unsuitable in this context and why measures are used instead.

---

> ### Comment · Reviewer_Fipq · 2024-11-24
> **Thanks for the reply.**
>
> Thanks authors for replying my comments and clarifying my confusion!
>
> I think this paper has good potential and I would encourage the authors to revise accoridng to the reply and other reviewer's comments. Particularly improve the accessibility of the paper and the numerical simulations.

---

> > ### Author Response · Authors · 2024-11-24
> >
> > Thank you for your constructive comments and for acknowledging the potential of our work. We greatly appreciate your feedback, which has been invaluable in improving our manuscript.
> >
> > In response to your suggestions and the feedback from other reviewers, we have revised the paper to enhance its accessibility and address the concerns raised. Specifically:
> >
> > 1. We have improved the clarity and explanations in the introduction, literature review, notation, and terms.
> > 2. We have added more examples for $\nu$ and $\mathbb{G}$.
> > 3. A more thorough simulation study is currently ongoing. Depending on progress, we aim to include the results either before the end of the discussion period or in the camera-ready version, should the paper be accepted.
> > To make it easier for reviewers to track the changes, all major updates have been highlighted in blue in the revised manuscript. These highlights will be reverted to black in the final version.
> >
> > Given the efforts to address the points raised and the potential of our work that you kindly highlighted, we would like to respectfully request that you reconsider your score, if possible. We believe the revisions and planned updates significantly strengthen the paper’s contribution and clarity.

---

> > > ### Comment · Reviewer_Fipq · 2024-11-26
> > > **Raising score to 5**
> > >
> > > Thanks for the revision of the paper. For the improved clarity, I raise the score to 5.

---

> > > > ### Author Response · Authors · 2024-11-26
> > > >
> > > > Thank you so much for your re-consideration.
> > > >
> > > > One point that we forgot to mention about only investigating g-formula is:
> > > >
> > > > In causal inference with longitudinal studies, the true causal effects are often not analytically computable due to data generating complexity (unlike in point exposure case one can hand compute the true average treatment effect). Instead, they are typically approximated numerically using methods like the g-computation formula through sampling, **the way we outlined in this paper**, with very large sample sizes (e.g., 𝑛=10^8), a standard practice for benchmarking estimator performance. My paper extends this approach to functional longitudinal data, demonstrating its applicability in more complex settings where direct computation is similarly infeasible.

---

### Author Response · Authors · 2024-11-18

A summary of revision:
1. More clarity and explanations on introduction, literature reviews, notation, terms;
2. More examples on $\nu$ and $\mathbb{G}$;
3. A more through simulation is ongoing, but depend on the progress it may happen either before the end of discussion period or camera ready version if accepted.

All major changes are highlighted in blue for reviewers to track changes easier and changed back to black fonts later.

---

### Author Response · Authors · 2024-11-26

A more complicated simulation including mortality and censoring, more covariates, a more complicated $\nu()$, requested by multiple reviewers, is now finished and added to the appendix.

---

### Meta-Review · Area_Chair_5Re6 · 2024-12-19

**Metareview:**

The paper provides steps towards causal adjustments with continuous-time data and situations with confounded feedback. While the discrete-time version is well-studied (including work by Robins dating back to the 80s), the continuous-time less so. The paper does provide advances towards this relevant problem, with an emphasis on a worked-out experimental example instead of a major benchmark - I think this was a good choice.

 It is fair to say that the novelty is relatively limited compared to recent progress, but on the other hand this line of work is not too well-known to the ICLR audience and bringing it to his audience that's a plus in that regard. The drawback is that even though terms will be familiar to experts in causal inference coming from a more statistical background, in general it is very dense for a more familiar ICLR audience.

**Additional Comments On Reviewer Discussion:**

Comments focused on clarity and novelty. I think the discussion was transparent, although there was a sense that clarity/novelty lies on the borderline.

---

### Decision · Program_Chairs · 2025-01-22

Accept (Poster)